# CD151 expression marks atrial- and ventricular-differentiation from human induced pluripotent stem cells

Misato Nakanishi-Koakutsu [1,2,3,4], Kenji Miki [1,5,6,10 ✉], Yuki Naka[1,2], Masako Sasaki[1,2], Takayuki Wakimizu[1,2], Stephanie C. Napier[2,7], Chikako Okubo [1], Megumi Narita [1], Misato Nishikawa[1], Reo Hata[1], Kazuhisa Chonabayashi[1,8], Akitsu Hotta [1,2], Kenichi Imahashi[2,7], Tomoyuki Nishimoto[2,9] & Yoshinori Yoshida [1,2 ✉]

Current differentiation protocols for human induced pluripotent stem cells (hiPSCs) produce heterogeneous cardiomyocytes (CMs). Although chamber-specific CM selection using cell surface antigens enhances biomedical applications, a cell surface marker that accurately distinguishes between hiPSC-derived atrial CMs (ACMs) and ventricular CMs (VCMs) has not yet been identified. We have developed an approach for obtaining functional hiPSC-ACMs and -VCMs based on CD151 expression. For ACM differentiation, we found that ACMs are enriched in the CD151$^{low}$ population and that CD151 expression is correlated with the expression of Notch4 and its ligands. Furthermore, Notch signaling inhibition followed by selecting the CD151$^{low}$ population during atrial differentiation leads to the highly efficient generation of ACMs as evidenced by gene expression and electrophysiology. In contrast, for VCM differentiation, VCMs exhibiting a ventricular-related gene signature and uniform action potentials are enriched in the CD151$^{high}$ population. Our findings enable the production of high-quality ACMs and VCMs appropriate for hiPSC-derived chamber-specific disease models and other applications.

[1] Center for iPS Cell Research and Application, Kyoto University, Kyoto 606-8507, Japan. [2] Takeda-CiRA Joint program (T-CiRA), Fujisawa 251-8555, Japan. [3] Division of Cardiology, Johns Hopkins University School of Medicine, Baltimore, MD 21205, USA. [4] Department of Surgery, Johns Hopkins School of Medicine, Baltimore, MD 21205, USA. [5] Center for Organ Engineering, Department of Surgery, Massachusetts General Hospital, Boston, MA 02114, USA. [6] Department of Surgery, Harvard Medical School, Boston, MA 02114, USA. [7] Global Advanced Platform, Takeda Pharmaceutical Company Limited, Fujisawa 251-8555, Japan. [8] Department of Hematology and Oncology, Graduate School of Medicine, Kyoto University, Kyoto 606-8507, Japan. [9] Orizuru Therapeutics Incorporated, Fujisawa 251-8555, Japan. [10]Present address: Premium Research Institute for Human Metaverse Medicine, Osaka University, Suita 565-0871, Japan. ✉email: kenjimiki.prime@osaka-u.ac.jp; yoshinor@cira.kyoto-u.ac.jp

Conventional methods used to differentiate human pluripotent stem cell-derived cardiomyocytes (hPSC-CMs) generate a mixture of different CM subtypes, including ventricular CMs (VCMs), atrial CMs (ACMs), and nodal-like cells[1–3]. Although such heterogeneity indicates the potential for producing an unlimited number of each subtype in vitro, it creates challenges for purifying chamber-specific cells for various applications such as disease modeling and drug testing.

Several groups have reported effective methods for differentiating hPSCs into each subtype. These methods mimic in vivo cardiac development by regulating several signaling pathways, such as WNT, BMP, Activin/Nodal, and retinoic acid (RA) signaling[4–6]. WNT signaling exerts a biphasic effect, promoting early cardiac development and inhibiting it later. Accordingly, appropriate inhibition of WNT signaling at the mesoderm stage leads to highly efficient differentiation of hPSC-CMs[7–10]. BMP signaling contributes to cardiac mesoderm differentiation in zebrafish, while Nodal signaling further promotes ventricular specification[11–13]. Embryos deficient in retinaldehyde dehydrogenase 2 (RALDH2), an enzyme involved in RA synthesis, show failed cardiac looping and defective atrial and sinus venosus development, thus indicating the importance of RA signaling for cardiovascular progenitor cell development[14–16]. RA signaling induces NR2F2 (COUP-TFII), a key transcriptional regulator, to direct the cell fate toward ACMs and directly regulates atrial- and ventricular-related genes involved in mouse cardiac development[17] and the differentiation of human embryonic stem cells into ACMs[5]. These findings are applicable to hPSC-CMs, where reduced BMP, Activin/Nodal, and RA signaling promote the generation of hPSC-ACMs (atrial-biased differentiation), while higher BMP and Activin/Nodal signaling favor the generation of hPSC-VCMs[18] (ventricular-biased differentiation). However, we noted the presence of undesired subtypes in each differentiation method (i.e., ventricular-like CMs were generated in atrial-biased differentiation and vice versa), implying that signal manipulation was insufficient for generating completely homogeneous subtype populations. In addition, studies have demonstrated differentiation methods using hPSCs with chamber-specific reporters[19–21] or surface proteins to enrich each subtype[6,22]. Although assays using surface markers help to select CM subtypes, we still lack the markers to effectively purify functionally homogeneous chamber-specific hPSC-CMs applicable to multiple cell lines.

In the present study, we identified CD151, a member of the tetraspanin family, as a potential marker of chamber-specific CMs derived from human induced pluripotent stem cells (hiPSCs). Using an atrial differentiation protocol, we showed that the expression of atrial genes was higher in CD151low ACMs than in CD151high ACMs. Moreover, whereas CD151high ACMs contained CMs with immature atrial action potentials (APs) and upregulated *NOTCH*-related genes, only CD151low ACMs showed atrial APs. We also revealed that Notch signaling inhibition enhanced atrial differentiation via *HEY2* suppression and that these ACMs exhibited significantly more functional properties than hiPSC-ACMs generated by the conventional protocol. Using a ventricular differentiation protocol, we demonstrated that the CD151high selection enriched for VCMs with high ventricular-related gene expression and activated mitosis to result in binucleation. These findings altogether indicate that CD151 can distinguish ACMs and VCMs derived from hiPSCs in a bivalent manner. Furthermore, we revealed a pivotal role of differential Notch signaling activities in CD151low and CD151high ACMs for atrial differentiation. Our study demonstrates that cardiac subtypes with molecular and functional properties of ACMs or VCMs can be efficiently obtained from hiPSCs-CMs.

## Results

### CD151 shows different expression patterns during hiPSC-ACMs and -VCMs differentiation.
We differentiated ACMs and VCMs from hiPSCs based on a previously reported method[6]. ACMs were generated with a low concentration of agonists for BMP4 (3 ng/mL) and Activin A (4 ng/mL) in the early stage of embryoid body (EB) formation (days 1–3), followed by treatment with a combination of WNT inhibitor (IWP-3), Nodal inhibitor (SB43152), VEGF, and 0.5–1 μM RA during cardiac mesoderm induction (days 3–6). We refer to this differentiation condition as atrial-inducing condition (AIC; Fig. 1a). For differentiation into VCMs, hiPSCs were treated with a high concentration of BMP4 (10 ng/mL) and Activin A (6 ng/mL) on days 1–3, followed by IWP-3, SB43152, VEGF, and BMP4 inhibitor (Dorsomorphin) on days 3–6, which we refer to as ventricular-inducing condition (VIC; Fig. 1a).

First, we utilized the TNNI1-EmGFP/TNNI3-mCherry double reporter hiPSC line generated in our previous study[23]. This line enables the visualization of CMs exhibiting green fluorescence due to the placement of EmGFP downstream of TNNI1 (Fig. 1b). In this study, we concentrated on the TNNI1-EmGFP marker (referred to as "TNNI-EmGFP" henceforth). On day 27, TNNI1+ cells in AIC- and VIC-EBs predominantly expressed MLC2a and MLC2v, respectively (Fig. 1c). We confirmed that AIC- and VIC-EBs also contained MLC2v- and MLC2a-positive CMs, respectively, suggesting that AIC and VIC resulted in a mixture of VCM-like and ACM-like cells. To determine whether the heterogeneity resulting from these differentiation protocols is reproducible, we differentiated and purified hiPSC-CMs from 1390C1, an independent hiPSC line, using the pan-CM marker SIRPA and other lineage markers (CD31, CD49a, CD90, and CD140b)[24] (Fig. S1a). Immunofluorescence analysis indicated that most AIC-CMs were positive for MLC2a but contained some MLC2v+ CMs, whereas VIC-CMs were predominantly positive for MLC2v but also contained MLC2a+ CMs (Fig. S1b). Despite long-term culture, MLC2v+ and MLC2a+ CMs were still present in day 60 AIC and VIC cultures, respectively (Fig. S1c). These observations indicated that even protocols with optimized concentrations of BMP4, Activin A, and RA for chamber-specific CM generation showed a mix of CM subtypes.

To enhance CM subtype specificity, we sought to identify differentially expressed markers present in AIC- and VIC-CMs. We performed screening of 212 cell surface markers with the panel of antibodies by comparing the expression patterns of CMs induced by AIC and VIC (Fig. S1d). We eliminated cell surface antigens whose expressions were affected only by BMP (3 ng/mL) and Activin A (4 ng/mL) without RA since RA signaling is critical for differentiation into ACMs. We found five cell surface antigens with different expression patterns between AIC- and VIC-CMs (Fig. S1e). Flow cytometry analysis showed that CD57, CD71, and CD98 were expressed in AIC and AIC without RA, thus disqualifying them as subtype-specific markers (Fig. S1e (i–iii)). Although the expression pattern of SSEA-4 in AIC was distinct from the other conditions (Fig. S1e iv), most AIC-CMs were SSEA-4 negative (Fig. S1f). Finally, CD151 was highly expressed in VIC- and AIC without RA-CMs but not in AIC-CMs (Fig. S1ev), and thus we focused on CD151 as a candidate marker to distinguish AIC-CMs from VIC-CMs.

We analyzed the expression levels of subtype-related genes to determine whether CD151 can separate ACMs from VCMs. We sorted CD151high/low CMs from AIC- and VIC-EBs (Fig. 1d) to analyze the expressions of atrial- and ventricular-related genes via qPCR. AIC-CD151low CMs (CD151low ACMs) highly expressed atrial-related genes, such as *NPPA, KCNA5*, and *KCNJ3*, compared to AIC-CD151high CMs (CD151high ACMs). In contrast, the expression levels of those genes in VIC-CMs were

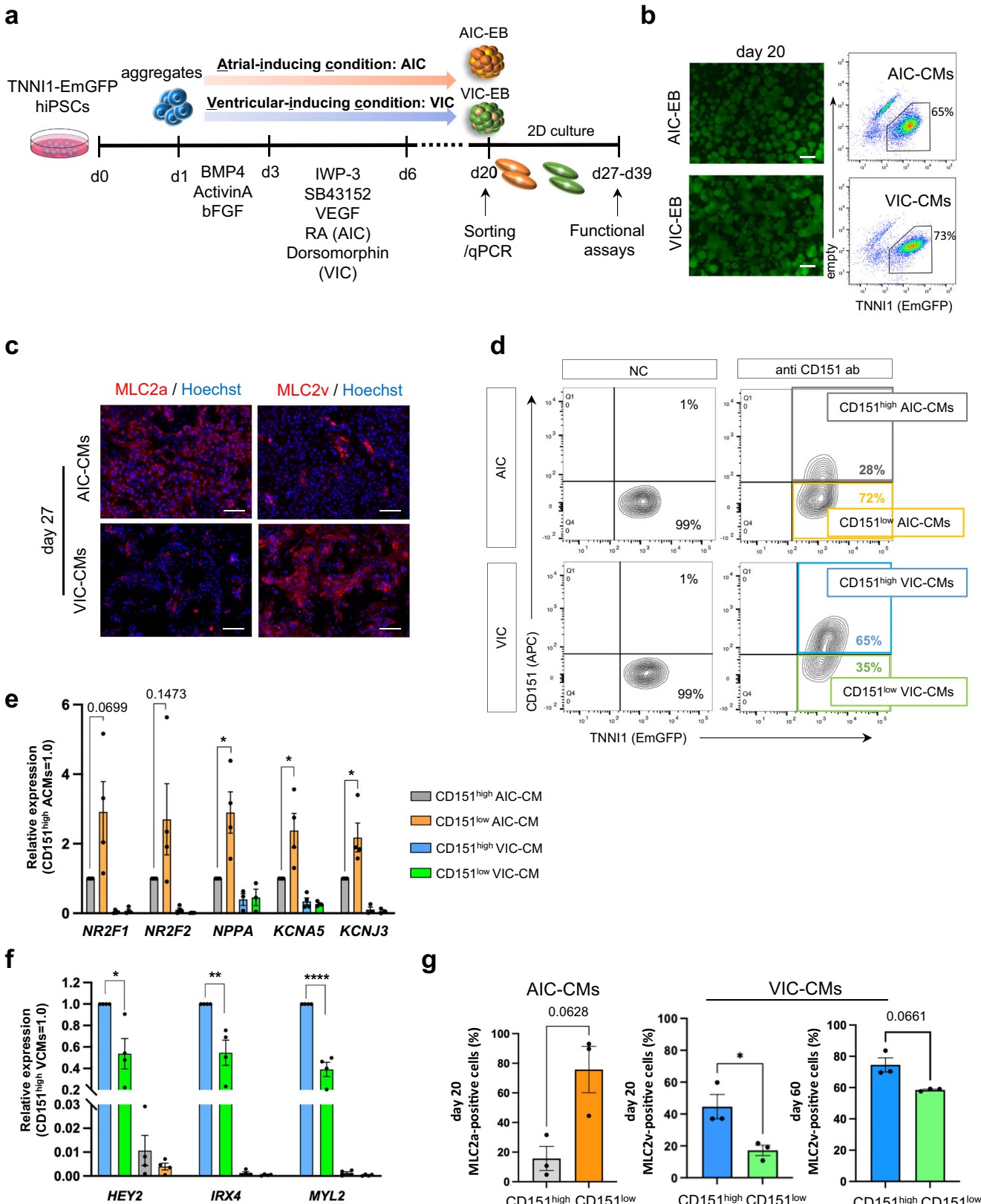

low regardless of CD151 selection (Fig. 1e). Furthermore, the expression levels of ventricular-related genes (*HEY2, IRX4,* and *MYL2*) in VIC-CD151[low] CMs (CD151[low] VCMs) or CD151[high/low] ACMs were significantly lower than those in VIC-CD151[high] CMs (CD151[high] VCMs) (Fig. 1f). To validate these findings in other hiPSC lines, we sorted CD151[high/low] CMs from AIC- and VIC-EBs using SIRPA and other lineage markers (Figs. S2a and S2d) and confirmed that atrial- and ventricular-related gene expression patterns in observed CD151[high/low] ACMs and VCMs were similar to those derived from TNNI1-EmGFP reporter hiPSCs (Figs. S2b, S2c, S2e, and S2f). On day 27, MLC2a-positive CMs were clearly detected among CD151[low]

**Fig. 1 Cell surface marker screening for hiPSC-ACMs and -VCMs. a** Schematic diagram and experimental schedule of cardiac differentiation protocol for AIC and VIC. **b** Representative fluorescence and flow cytometry images of day 20 AIC- and VIC-CMs. Scale bars = 100 μm. (See also Fig. S1a). **c** Representative immunofluorescence images showing MLC2a, MLC2v, and Hoechst staining on day-27 AIC- and VIC-CMs. Scale bars = 100 μm. (See also Fig. S1b). **d** Representative flow cytometry images of day 20 AIC- and VIC-CMs stained with CD151 antibody. The cut-off between CD151$^{high}$ and CD151$^{low}$ cells was determined using unstained CMs as a negative control. The gate containing most negative control CMs (>99%) was set as the CD151$^{low}$ gate (See also Figs. S2a and S2d). **e** Relative expression of atrial-related genes in CD151$^{high/low}$ AIC- and VIC-CMs, compared to CD151$^{high}$ AIC-CMs. $n = 4$ independent differentiation experiments per group. Data are expressed as mean ± SEM. Statistical analysis was conducted between CD151$^{high}$ AIC-CM and CD151$^{low}$ AIC-CM using an unpaired two-tailed $t$ test. *$p < 0.05$. (See also Figs. S2b and S2e). **f** Relative expression of ventricular-related genes in CD151$^{high/low}$ VIC- and AIC-CMs compared to that of CD151$^{high}$ VCMs. $n = 4$ independent differentiation experiments per group. Data are expressed as mean ± SEM. Statistical analysis was conducted between CD151$^{high}$ VIC-CM and CD151$^{low}$ VIC-CM using an unpaired two-tailed $t$ test. *$p < 0.05$, **$p < 0.01$, and ****$p < 0.0001$ (See also Figs.S2c and S2f). **g** MLC2a-positive cells fraction (%) in CD151$^{high/low}$ AIC-CMs on day 20 (left). MLC2v-positive cells fraction (%) in CD151$^{high/low}$ VIC-CMs on day 20 and day 60 (right). $n = 3$ independent differentiation experiments per group. Data are expressed as mean ± SEM. Statistical analysis was conducted using an unpaired two-tailed $t$ test. *$p < 0.05$.

ACMs. Whereas CD151$^{low}$ VCMs showed some MLC2a-positive cells, CD151$^{high}$ VCMs did not (Fig. S2g). Furthermore, we detected 93% and 36% of MLC2a-positive cells in CD151$^{low}$ - and CD151$^{high}$ AIC-CMs, respectively by flow cytometric analysis on day-20 TNNI1-EmGFP hiPSCs, and 45% and 17% of MLC2v-positive cells in CD151$^{high}$- and CD151$^{low}$ VIC-CMs, respectively (Fig.1g). These findings suggest that CD151$^{low}$ AIC-CMs and CD151$^{high}$ VIC-CMs were enriched ACMs and VCMs, respectively.

**CD151$^{high/low}$ CMs exhibited differential electrophysiological properties**. Next, we investigated the electrophysiological properties of CD151$^{high/low}$ ACMs and VCMs by measuring action potentials (APs) using a patch-clamp method (Fig. 2a). In AIC, CD151$^{high}$ ACMs showed a significantly slower maximum upstroke velocity (Vmax) than CD151$^{low}$ ACMs, which is a characteristic feature of nodal-like CMs (Fig. 2b). Furthermore, although there was no significant difference between the maximal depolarization potentials (MDPs) of CD151$^{high}$ ACMs and CD151$^{low}$ ACMs, the action potential amplitude (APA) of CD151$^{low}$ ACMs was increased compared to that of CD151$^{high}$ ACMs (Fig. 2b). To determine the fraction of cells showing atrial type APs (AP duration at 30% of repolarization (APD30) / AP duration at 90% of repolarization (APD90) < 0.3 and Vmax > 10), we classified CD151$^{high/low}$ cells into each CM subtype (ventricular-type - APD30/90 > 0.3 and Vmax > 10; nodal-type - Vmax ≤ 10). While no atrial type APs and 81% of nodal-like CMs were detected in CD151$^{high}$ ACMs, ACMs exhibiting atrial type APs were found only in CD151$^{low}$ population, and they exhibited significantly smaller APD30/90 than CD151$^{high}$ VCMs (Figs. 2c and S3b). Although the proportion of atrial-type APs was only 35% (Fig. 2c and Supplementary Table 1), these data show that ACMs were enriched in CD151$^{low}$ ACMs, consistent with the high expression of atrial-related genes in CD151$^{low}$ populations (Figs.1e, S2b, and S2e).

Although the levels of Vmax, APA, and MDP were similar in both CD151$^{high}$ and CD151$^{low}$ VCMs under VIC (Fig. S3a), APD30 and APD50 were significantly shorter in CD151$^{low}$ VCMs than CD151$^{high}$ VCMs (Fig. 2d). Additionally, we found that while CD151$^{low}$ VCMs contained cells showing atrial-type APs (22%), CD151$^{high}$ VCMs only displayed ventricular-type APs (93%) (Fig. 2e; Supplementary Table 1), consistent with the shorter APDs seen in CD151$^{low}$ VCMs (Fig. 2d). These data demonstrate that hiPSC-VCMs were purified by CD151 expression.

**CD151 does not directly regulate subtype differentiation**. To determine whether CD151 directly regulates atrial and ventricular differentiation, we deleted CD151 from the TNNI1-EGFP hiPSC line using CRISPR-Cas9. We sorted CD151 negative cells from

gRNA-transfected cells (Fig. 3a) and confirmed that CD151 was not expressed in generated knockout (KO) hiPSC line (CD151 KO hiPSCs) at the protein level (Figs. 3b and S4a). Genomic validation revealed that a frameshift was induced with 100% efficiency in CD151 KO iPSCs (Fig. S4b). Furthermore, there were no mutations at the predicted off-target sites in exons (Fig.S4c). Subsequently, we cultured CD151 KO hiPSCs and differentiated them into the AIC- and VIC-CMs. Both CMs did not express CD151 at the protein level (Figs. 3b and S4d). Using this CD151 KO line, we analyzed the expression of atrial and ventricular marker genes in AIC- and VIC-CMs on day 20, respectively, and compared the expression levels with those in WT CMs. No significant changes in atrial (Fig. 3c) or ventricular (Fig. 3d) marker gene expression were observed in CD151 KO cells, indicating that CD151 does not regulate the atrial and ventricular differentiation.

**Notch signaling inhibition promotes atrial differentiation from hiPSCs**. Next, we investigate whether CD151 expression is reflected by difference in signaling molecules and pathways regulating subtype. To examine this possibility in detail, we performed RNA sequencing (RNA-seq) of CD151$^{high/low}$ ACMs and VCMs. In principal component analysis (PCA), the first component (PC1), which accounted for 55.99% of the variance in gene expression, separated AIC- and VIC-CMs, and the second component (PC2, 13.41% of the variance) separated the expression levels of CD151 in AIC- and VIC-CMs (Fig. S5a). We performed GO and Reactome pathway analyses with highly correlated PC2 (|factor loadings of PC2| > 0.8) in differentially expressed genes (DEGs) between CD151$^{high}$ ACMs and CD151$^{low}$ ACMs (Supplementary Data 1). GO enrichment analysis and hierarchical clustering revealed an enrichment of cell adhesion-related genes (Figs. S5b and S5c). Additionally, Reactome pathway analysis revealed a prominent molecular signature by the Notch signaling pathway (Fig. 4a). The expression of NOTCH4 and its ligands (DLL4 and DLK1) made a high contribution to PC2 (loadings were 0.84, 0.90, and 0.90, respectively) (Fig. S5d), suggesting that the expression of NOTCH4, DLL4, and DLK1 was highly correlated with CD151 expression. To validate our findings, we examined the expression levels of NOTCH4 and DLL4 in CD151$^{high/low}$ ACMs via qPCR and found that their expression levels in CD151$^{high}$ ACMs were significantly higher than those in CD151$^{low}$ ACMs (Fig. 4b).

Notch signaling is regulated via direct cell-cell interactions between cells expressing the ligand and cells expressing the Notch receptor. Therefore, based on our findings, we hypothesized that Notch signaling activated by cell-cell interactions in CD151$^{high}$ ACMs negatively regulated atrial differentiation in the AIC. To determine whether Notch signaling inhibits atrial differentiation under AIC, we treated AIC-EBs with a gamma-secretase

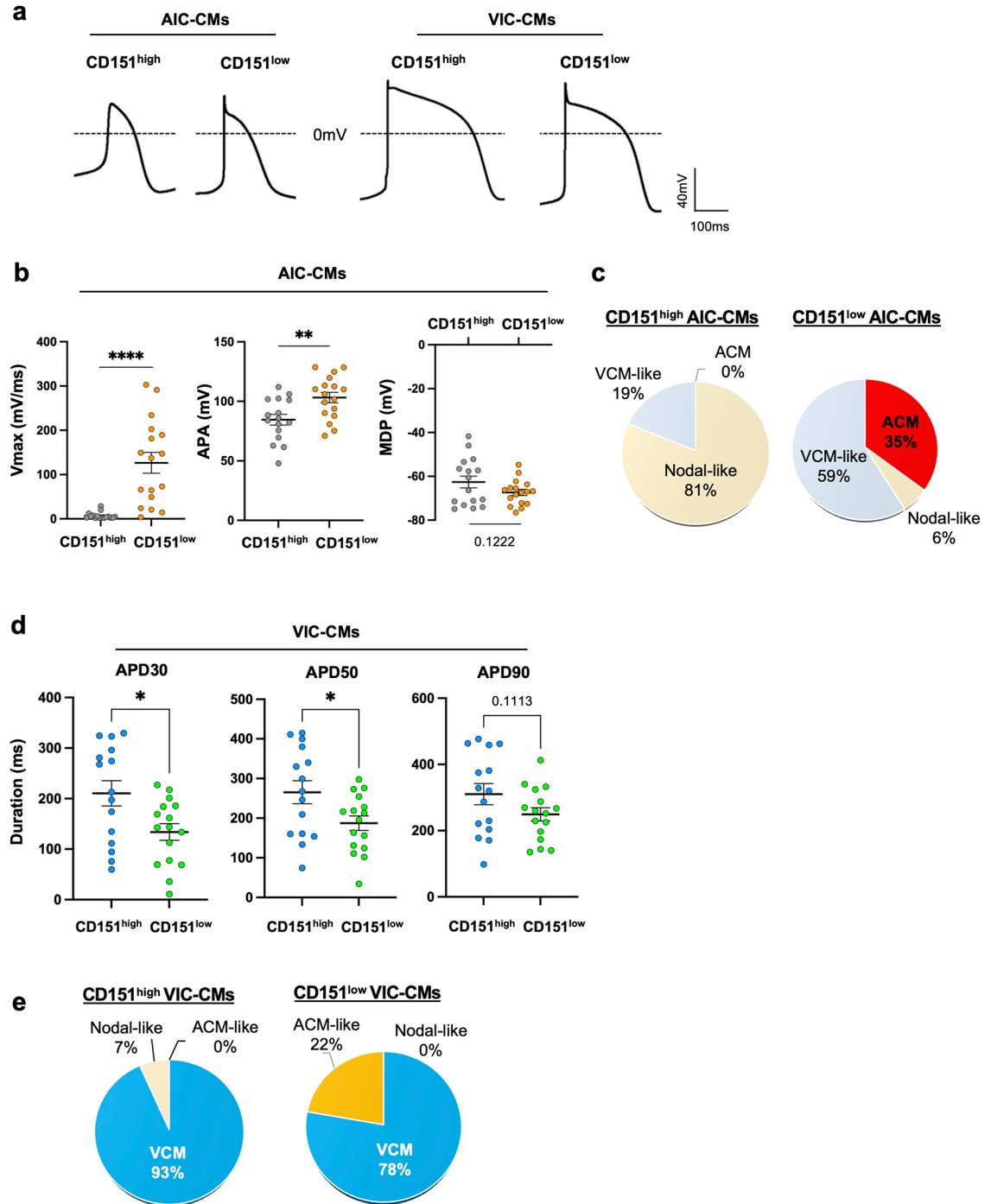

**Fig. 2 Electrophysiological properties of CD151$^{high/low}$ ACMs and VCMs. a** Representative APs recorded in CD151$^{high/low}$ ACMs from AIC-CMs and CD151$^{high/low}$ VCMs from VIC-CMs. **b** Electrophysiology measurements of Vmax, APA, and MDP in CD151$^{high}$ ACMs ($n = 16$) and CD151$^{low}$ ACMs ($n = 17$) from three independent differentiation experiments. Data are presented as mean ± SEM. Statistical analysis was conducted using an unpaired two-tailed $t$ test. **$p < 0.01$ and ****$p < 0.0001$. **c** The proportions of AIC-CMs with atrial (ACM; APD30/90 < 0.3, Vmax > 10), ventricular-like (VCM-like; APD30/90 ≥ 0.3, Vmax > 10), and nodal-like APs (Vmax ≤ 10) in CD151$^{high}$ ACMs ($n = 16$) and CD151$^{low}$ ACMs ($n = 17$). **d** Electrophysiology measurements of APD30, APD50, and APD90 in CD151$^{high}$ VCMs ($n = 15$) and CD151$^{low}$ VCMs ($n = 16$) from three independent differentiation experiments. Data are presented as mean ± SEM. Statistical analysis was conducted using an unpaired two-tailed $t$ test. *$p < 0.05$. (See also Fig. S3b). **e** The proportions of VIC-CMs with ventricular (VCM), atrial-like (ACM-like), and nodal-like APs in CD151$^{high}$ ($n = 15$) and CD151$^{low}$ VCMs ($n = 18$). The criteria are described in (**c**).

inhibitor, LY411575, on days 8–20 of differentiation (Fig. 4c) because *NOTCH4* is highly expressed in AIC-EBs after day 8 (Fig. S5e). On day 20, the ratio of CD151$^{high}$ ACMs to CD151$^{low}$ ACMs remained unchanged with or without LY411575 treatment (Fig. S5f), suggesting that CD151 expression is not regulated by Notch signaling. We sorted TNNI1$^+$ CMs and then analyzed the expression of subtype marker genes. The qPCR results showed that LY411575 treatment increased the expression of atrial-related genes in both CD151$^{high}$ and CD151$^{low}$ ACMs (Figs. 4d and e), confirming that the inhibition of Notch signaling

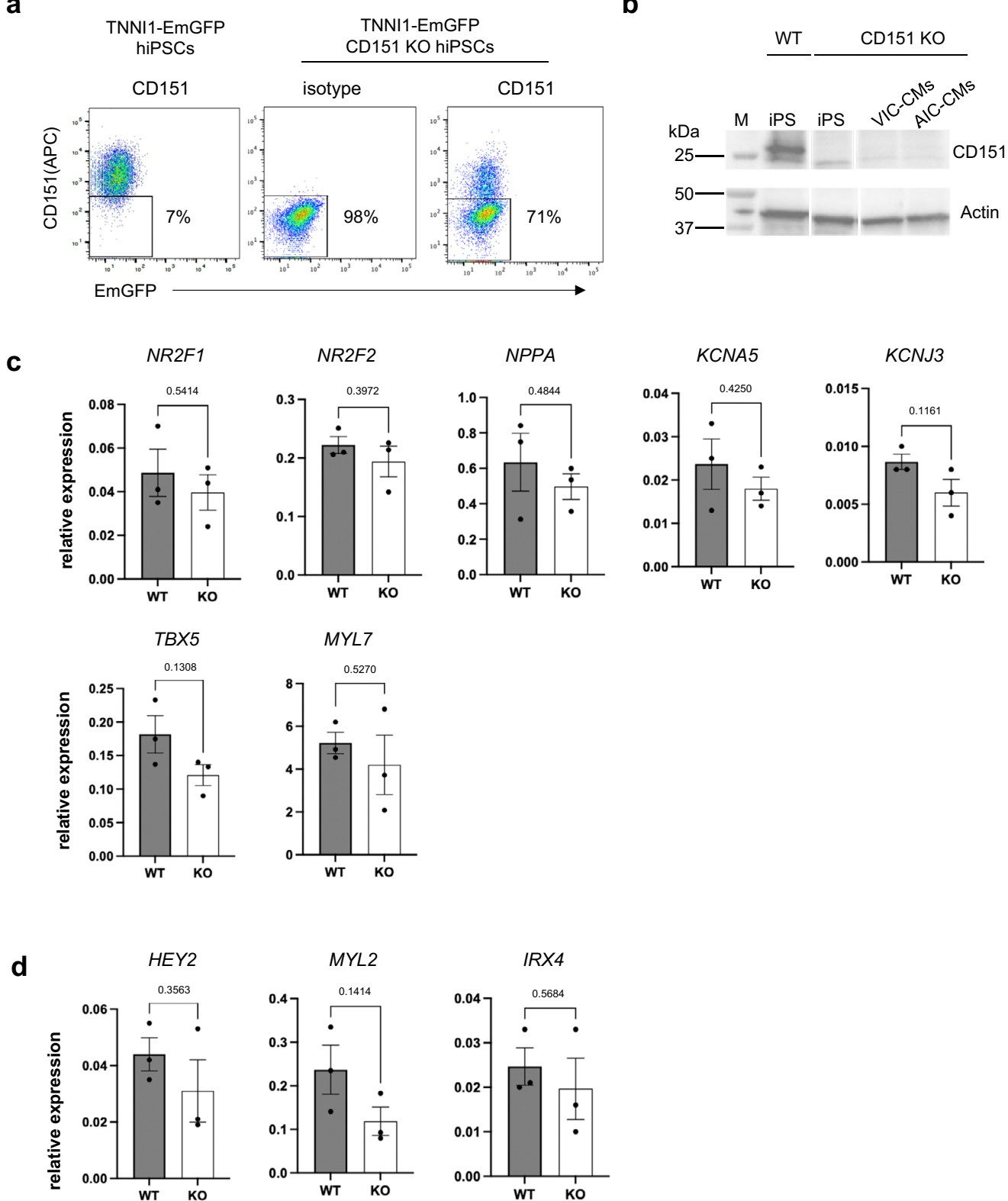

**Fig. 3 CD151 KO hiPSC-derived AIC-CMs and VIC-CMs. a** Generation of the CD151 KO hiPSC line. Representative FACS plot of Cas9 alone-transfected hiPSCs (left), CD151 KO gRNA-transfected hiPSCs with the isotype control (middle) and CD151 stained sample (right). About 71% of the transfected cells were overlapped with the isotype control. This population was sorted and harvested. **b** Western blots analysis of CD151 expression. The samples were WT-hiPSCs, CD151 KO gRNA transfected-hiPSCs, VIC-CMs, and AIC-CMs. M shows the molecular weight marker. Top panel: anti-CD151 antibody; bottom panel: anti-Actin as loading control. The sample lanes were re-arranged of non-adjacent lanes in the original blot images (See also Fig. S4a). Atrial marker expressions in WT and CD151 KO hiPSC-derived AIC-CMs (**c**) and ventricular marker expressions (**d**) in WT and CD151 KO hiPSC-derived VIC-CMs. $n = 3$ independent differentiation experiments per group. Data are expressed as mean ± SEM. Statistical analysis was conducted using an unpaired two-tailed $t$ test.

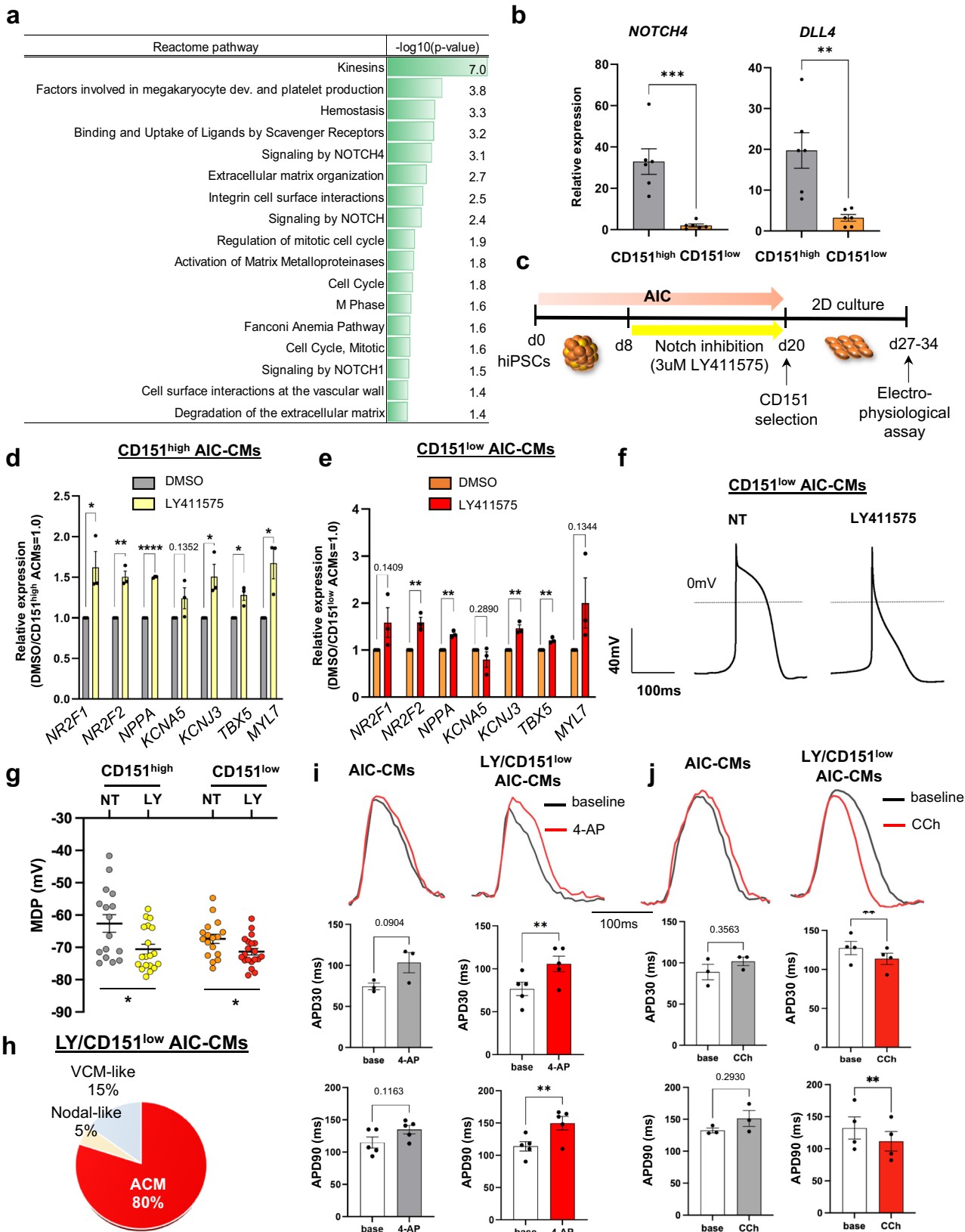

promotes atrial differentiation in AIC-EBs. Moreover, *HEY2*, a known target of the Notch signaling pathway and a suppressor of atrial-related gene expression[25], was downregulated by LY411575 (Fig. S5g). Although the causal role of HEY2 in atrial differentiation requires further investigation, our data highlight the potential involvement of HEY2 in mediating the gene

expression changes downstream of Notch signaling, the inhibition of which promotes atrial specification.

To clarify whether Notch inhibition contributes to atrial differentiation in CD151[high/low] ACMs, we analyzed APs using patch-clamp experiments. The waveforms of LY411575/ CD151[low] ACMs showed no plateau phase (Fig. 4f), and MDPs

**Fig. 4 Notch signaling inhibition promotes atrial differentiation from hiPSCs. a** Reactome pathway analysis in AIC-CMs. Three Notch signaling pathways were included in the 17 pathways with a significance level of $p < 0.05$. **b** Relative expression of *NOTCH4* and *DLL4* genes in CD151[high] ACMs compared to CD151[low] ACMs. $n = 6$ independent differentiation experiments per group. Data are presented as mean ± SEM. Statistical analysis was conducted using an unpaired two-tailed *t* test (See also Fig. S5d). **$p < 0.01$ and ***$p < 0.001$. **c** Schematic diagram of the assay schedule for Notch inhibition treatment for AIC differentiation. **d** Relative expression of atrial-related genes in CD151[high] ACMs differentiated with LY411575 or DMSO as vehicle control. $n = 3$ independent differentiation experiments per group. Data are presented as mean ± SEM. Statistical analysis compared two treatments for each gene using an unpaired two-tailed *t* test. *$p < 0.05$, **$p < 0.01$, and ****$p < 0.0001$. **e** Relative expression of atrial-related genes in CD151[low] ACMs differentiated using LY411575 or DMSO as vehicle control. $n = 3$ independent differentiation experiments per group. Data are expressed as mean ± SEM. Statistical analysis compared two treatments for each gene using an unpaired two-tailed *t* test. **$p < 0.01$. **f** Representative AP waveforms of CD151[low] ACMs differentiated with vehicle (DMSO) or LY411575 treatment. **g** Electrophysiological measurements of MDP in CD151[high/low] ACMs differentiated with vehicle or LY411575 treatment. For CD151[high] ACMs, DMSO ($n = 16$) and LY411575 treatment ($n = 19$); for CD151[low] ACMs, DMSO ($n = 17$) and LY411575 treatment ($n = 20$). Data are presented as mean ± SEM. Statistical analysis was conducted using an unpaired two-tailed *t* test. *$p < 0.05$. **h** The proportion of atrial (ACM), VCM-like, and nodal-like APs in CD151[low] ACMs differentiated with LY411575. **i** Representative APs of AIC-CMs and LY/CD151[low] ACMs detected using FluoVolt dye before and after 4-AP treatment or (**j**) CCh treatment. APD30 and APD90 were analyzed using the averages of 10 APs at baseline and after drug treatment under 4 Hz pacing. $n = 3–5$ independent differentiation experiments. Data are expressed as mean ± SEM. Statistical analysis was conducted using a paired two-tailed *t* test. **$p < 0.01$.

were significantly more hyperpolarized than those of the untreated groups (Fig. 4g). APAs and Vmax of LY411575/CD151[high] ACMs were significantly increased (Figs. S5h and S5i). Furthermore, 80% of the LY411575/CD151[low] ACMs were found to exhibit atrial-type APs (Fig. 4h). These results demonstrated that ACMs were generated and enriched with high efficiency in the CD151[low] ACMs treated with LY41157. Finally, to verify the functional properties of LY411575/CD151[low] ACMs, the drug responsiveness of CM sheets was measured with the atrial-specific ultra-rapid outward current ($I_{Kur}$) blocker, 4-aminopyridine (4-AP), and acetylcholine-induced current ($I_{Ach}$) agonist, carbachol (CCh), using a membrane potential-sensitive fluorescent dye (FluoVolt). To reveal the practical advantage of combining CD151 selection following Notch inhibition, we compared the responsiveness LY411575/CD151[low] ACMs and conventional AIC-CMs. LY411575/CD151[low] ACMs showed significantly prolonged and notably shortened APD following 4-AP (Fig. 4i) and CCh treatments (Fig. 4j), respectively. In contrast, AIC-generated cells did not respond to these reagents (Figs. 4i and j). These data indicated that LY411575/CD151[low] ACMs were more functional as ACMs compared to conventionally differentiated ACMs. Altogether, these findings suggest that Notch signaling contributes to atrial differentiation potentially by regulating *HEY2* expression (Fig. S5j), and an approach combining Notch inhibition (which promotes the differentiation of ACMs) and CD151-based selection significantly improves the purification of functional ACMs.

**CD151[high] VCM gene expression profile and binucleation are indicative of advanced differentiation**. Under VIC, we demonstrated that CD151[high] VCMs are a purified population of VCMs without contamination by CMs exhibiting atrial-type APs. To identify the gene signatures of VCM differentiation dependent on CD151 expression, we analyzed RNA-seq data and identified *MYL2* and *HEY2* as DEGs in VIC-CMs (Supplementary Data 2). To determine whether CD151[high] VCMs express higher levels of other ventricular-related genes, we analyzed the expression of eight ventricular marker genes expressed in human ventricular tissues[26]. Six genes (*MYH7, MYL3, MYL2, TNNC1, PCDH7,* and *SMYD2*) were significantly upregulated in CD151[high] VCMs (Fig. 5a). Next, we performed GO and Reactome pathway analyses, which highlighted cell cycle and mitotic pathways (Figs. S6a and S6b). During human heart development, CMs stop proliferating after birth, and nuclear division occurs in postnatal CMs without cytokinesis, leading to the binucleation of CMs. To investigate whether mitotic progression in CD151[high] VCMs contributes to the proliferation or binucleation of CMs, we

analyzed the expression of Ki-67, a proliferation marker. Flow cytometric analysis revealed that the percentages of Ki-67[+] cells in CD151[high] VCMs and CD151[low] VCMs were 4.5% ± 0.9% and 5.8% ± 2.1%, respectively, indicating no significant difference between the two groups (Figs. 5b and c). Next, we investigated if CD151[high] VCMs contained more binuclear cells because binucleated CMs are found in adult mice and as human hearts mature[27,28]. Flow cytometric analysis using Hoechst revealed that CD151[high] VCMs contained significantly more binucleated CMs (11.6% ± 3.0%) than CD151[low] VCMs (2.7% ± 0.4%) (Figs. 5b and 5d). Increasing binucleated CMs without cell proliferation was also observed using another hiPSC line (Figs. S6c and S6d). These results indicated that CD151[high] VCMs contain more binuclear CMs, accounting for the activation of cell cycle- and mitosis-related genes in CD151[high] VCMs. Considered together, CD151[high] VCMs are a population undergoing advanced VCM differentiation.

## Discussion
In this study, we demonstrated hiPSC-ACMs to enrich in the CD151[low] population of AIC-CMs, and conversely, the CD151[high] VIC-CM population to contain enriched hiPSC-VCMs. Moreover, we identified CD151 as an indicator of Notch signaling and showed for the first time that its inhibition enhances atrial differentiation from hiPSCs. Under VIC, CD151[high] VCMs were purified binuclear VCMs with high ventricular genes expression, while CD151[low] VCMs exhibited low ventricular gene expression and contained some atrial-like CMs.

Chamber-specific differentiation protocols[5,6], the identification of chamber-specific markers[6,22], and the generation of chamber-specific reporter hPSCs[19–21] have been used to acquire each CM subtype selectively. However, current differentiation protocols, including the conventional AIC and VIC we used here, produce heterogeneous populations of hiPSC-CMs with varying degrees of maturity and quality. Although purification methods that use chamber-specific reporter hPSCs are very efficient, they cannot be applied widely to non-reporter cell lines. Veevers et al. showed that although hESC-VCMs were enriched in CD77[+]/CD200[-] population, CD77 expression depended on the cell line used[22]. Indeed, CD77 did not react with hiPSC-CMs in our screening panel (Fig. S1d). Here, we found that CD151, in contrast, is a unique and robust marker of chamber specification in both ACMs and VCMs, and its differential expression can be applied to isolate functional CM subtypes derived from multiple hiPSC lines. We demonstrated that CD151 can also enrich the CD151[low] AIC-CM population that exhibits a high expression of atrial genes and the CD151[high] VIC-CMs, characterized by a high expression

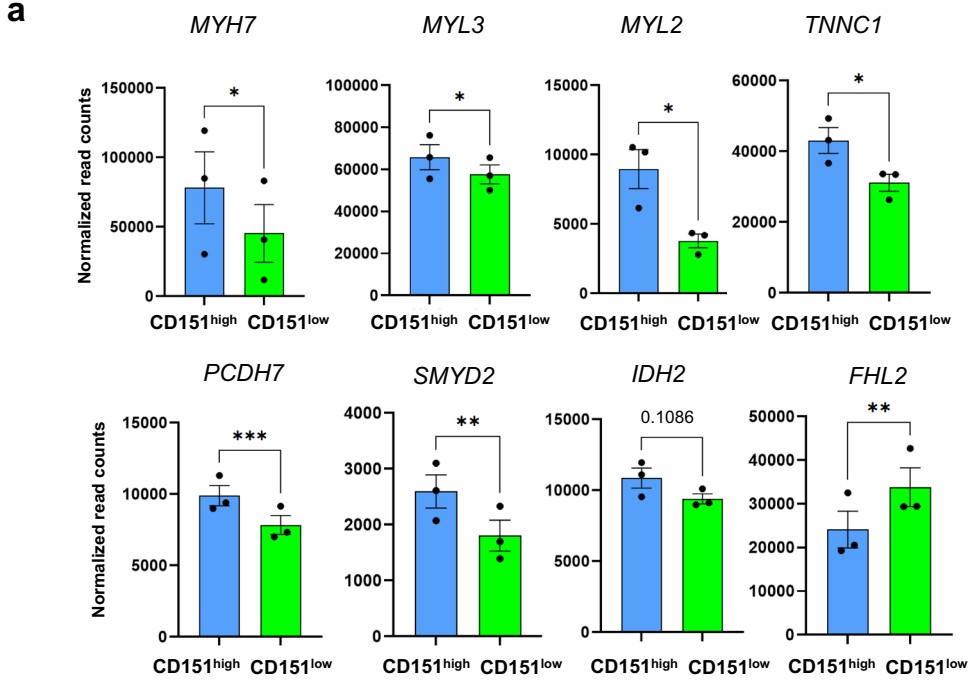

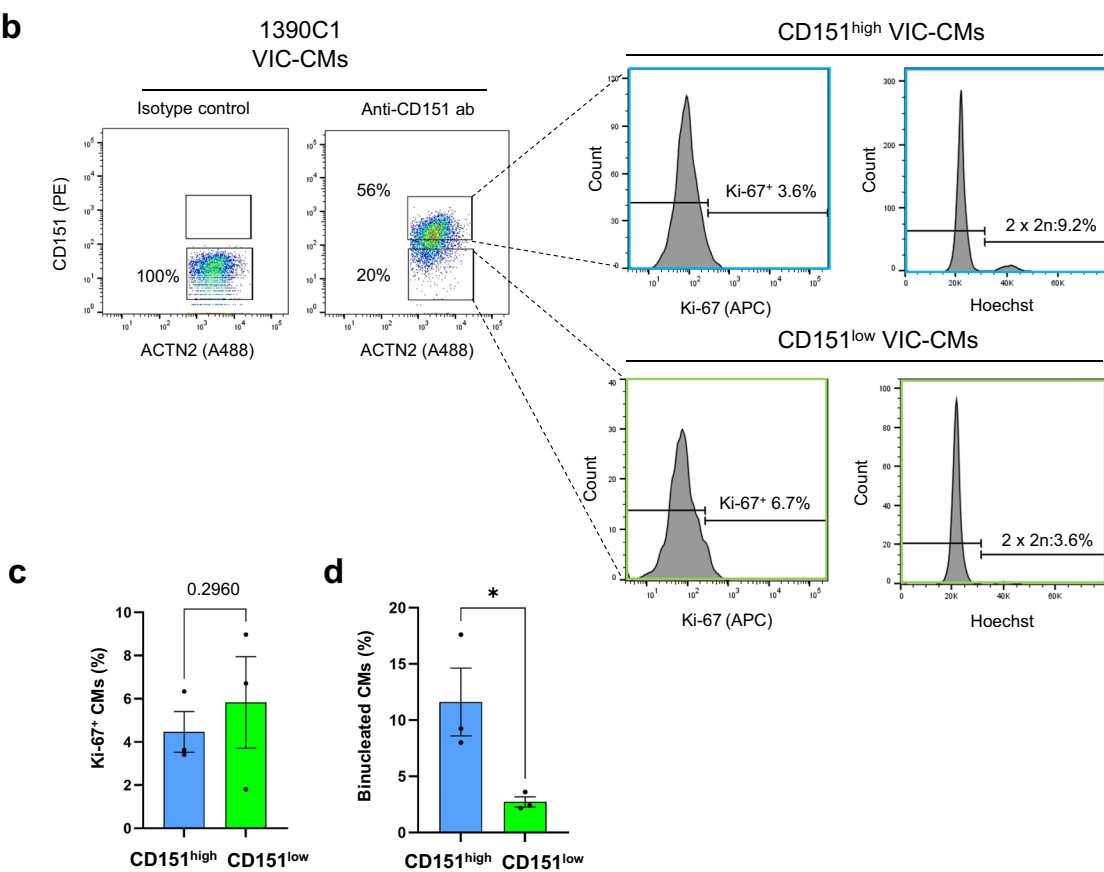

of ventricular genes, in monolayer differentiation (Figs. S7a and S7b). Our approach may help advance the biomedical applications of hiPSC-CMs.

CD151 belongs to the tetraspanin superfamily, which is involved in cellular processes, such as cell differentiation, proliferation, adhesion, migration, and intracellular signaling via various membrane proteins[29]. The known functions of CD151 have been studied primarily in oncology relating to invasion and metastasis. However, its role in cardiac development, differentiation, and chamber specification is completely unknown. We revealed a difference in CD151 expression between hiPSC-ACMs and -VCMs at 3–4 weeks post-differentiation. We also found that

**Fig. 5 CD151^high VCMs expressed higher levels of ventricular genes and contained binuclear CMs. a** The expression levels of eight ventricular marker genes in CD151^high/low VCMs. Data represent normalized read counts from RNA-seq data. $n = 3$ independent experiments per group. The data are expressed as mean ± SEM. Statistical analysis was conducted using an unpaired two-tailed $t$ test. *$p < 0.05$, **$p < 0.01$, and ***$p < 0.001$. **b** Representative flow cytometry images of day-20 VIC-CMs stained with ACTN2, CD151, Ki-67, and Hoechst derived from the 1390C1 hiPSC line. **c** The percentage of Ki-67^+ cells in CD151^high VCMs and CD151^low VCMs derived from the 1390C1 hiPSC line. $n = 3$ independent experiments per group. Data are expressed as mean ± SEM. Statistical analysis was conducted using an unpaired two-tailed $t$ test. **d** Percentage of binuclear CMs in CD151^high VCMs and CD151^low VCMs derived from the 1390C1 hiPSC line. $n = 3$ independent differentiation experiments per group. Data are expressed as mean ± SEM. Statistical analysis was conducted using an unpaired two-tailed $t$ test. *$p < 0.05$.

*CD151* expression was upregulated in both AIC- and VIC-CMs as differentiation progressed (Fig. S7c). Consistent with this, the ratios of CD151^high to CD151^low population increased in a time-dependent manner in both AIC- and VIC-CMs (Fig. S7d). To investigate whether in vivo atrial and ventricular tissues show a similar difference in *CD151* expression during development, we analyzed *CD151* expression at the single-cell level in mice and human left atrium (LA) and ventricle (LV) using published data[30,31]. In line with our in vitro results, *CD151* expressed higher in LV than in LA at mouse E10.5, human 7w, and 13w. However, such expression differences diminished during development (Fig. S7e). We also examined whether CD151^low AIC-CMs and CD151^high VIC-CMs could maintain their subtype phenotypes during extended culture. On day 60, no significant disparity in the expression of atrial marker genes was observed between high/low cells in AIC-CMs. Taken together, these findings indicate that CD151 is a transient marker of CM differentiation corresponding to embryonic heart organogenesis, and suggest that CD151 might serve as a marker of maturation level in AIC-CMs around day 20; CD151^low AIC-CMs were more mature than CD151^high AIC-CMs. The capacity to select more matured cells at an early stage will hold significant promise for future applications in disease models. In contrast, regarding VIC-CMs, while there was no difference in *MYL2* expression between CD151 high/low cells on day 60, *HEY2* and *IRX4* expression levels were still reduced in CD151^low VIC-CMs compared to CD151^high VIC-CMs (Fig. S7f). Since *MYL2* expression increased during maturation, this suggests that CD151 may mark maturation state, at the same time, CD151^high and CD151^low VIC-CMs may alternatively represent cell subpopulations with distinct molecular characteristics. Due to their high expression of ventricular marker genes, CD151^high VIC-CMs are a more suitable choice for further investigation for disease modeling and drug testing.

Integrin is a family of proteins reported previously to interact with CD151. Transcriptome analysis showed that *ITGA6* expression in AIC-CMs was higher than in VIC-CMs (Fig. S7g) and was inversely correlated with CD151 expression under AIC. These results substantiate a report that distinguished mouse ACMs by their ITGA6 expression[32]. However, flow cytometry using the CD49f (ITGA6) antibody indicated that CD49f did not help to differentiate between VIC- and AIC-CMs at the protein level because most VIC- and AIC-CMs were CD49f-positive (Fig. S7h). Our screening showed that two other tetraspanins, CD63 and CD81, were expressed in TNNI1-EmGFP reporter hiPSC-CMs, albeit at lower levels than that of CD151, but their levels were similar between AIC- and VIC-CMs (Fig. S1d). These results suggest CD151-specific signaling may be activated in AIC- and VIC-CM differentiation.

In this study, we found that *NOTCH4* and its ligands, *DLL4* and *DLK1*, were highly correlated with the expression of CD151 in AIC-CMs. Notch signaling is activated by cell-cell adhesion due to the binding of Notch receptors to their ligands. Hierarchical clustering analysis revealed that the GO term enriched by DEGs in CD151^high ACMs was cell-cell adhesion (Fig. S5c). These findings suggest that CD151 expression is associated with

Notch signaling activation induced by cell-cell adhesion. Although the Notch signaling pathway is highly conserved in multicellular organisms and regulates cell fate decisions in various differentiation processes, little is known about its role in atrial differentiation from hPSCs. In the present study, CD151^high ACMs expressed higher levels of *NOTCH4* and *DLL4* than CD151^low ACMs and showed nodal-like APs with slow Vmax in patch-clamp experiments. To determine if CD151^high ACMs are similar to sinoatrial node (SAN) CMs, we examined the expression levels of SAN-specific genes and found no changes in CD151^high ACMs (Fig. S7i), indicating CD151^high ACMs were molecularly distinct from SAN CMs. Moreover, a previous study reported that the APs of immature hiPSC-derived CMs resemble those of SAN CMs based on the expression of the funny current ($I_f$) channel and low levels of the inward rectifier potassium current ($I_{k1}$) channel[33]. Thus, these results collectively suggested that CD151^high ACMs were immature ACMs with high Notch signaling activity. A previous report showed that transient Notch activation in hPSC-derived cardiac mesoderm promotes immature cardiomyocytes with no change in their subtypes[34]. Though the time points of Notch activation or inhibition are different between that report and our study, Notch signaling may be critical for regulating the maturation of both VCMs and ACMs rather than cell fate.

Furthermore, Notch signaling inhibition with LY411575 increased the number of ACMs showing atrial-type AP waveforms and elevated expression of atrial-related genes accompanied by reduced *HEY2* expression (Figs. 4d–f, and S5g). These results indicate the inhibition of Notch signaling has the potential to promote atrial differentiation by reducing the expression of *HEY2*, a downstream target of Notch signaling and a repressor of atrial-related genes, such as *TBX5, NPPA,* and *MYL7*[25] (Fig. S5j). In contrast, Notch activation after cell fate determination may not contribute to VCM differentiation efficiency since Notch signaling-related genes were expressed lowly in VIC-CMs (Fig. S5d).

Under VIC, we demonstrated that whereas VCMs express higher levels of ventricular marker genes and contain more binuclear cells when enriched in the CD151^high VCM population, the CD151^low VCM population had many atrial-like CMs (22%), potentially due to insufficient *HEY2* expression to prevent ventricular differentiation in CD151^low cells. Furthermore, multinucleated CMs constitute approximately 20–30% of CMs in the human heart[28], and there is a substantial difference in binucleation levels exists between VCMs and ACMs, as shown by the relative proportion of binuclear CMs in the mouse ventricle and atria, at approximately 80% and 14%, respectively[35]. Altogether, although species differences must be taken into consideration, the binucleation of CMs during development may be a prominent feature of VCMs.

In conclusion, we demonstrated that CD151 expression is correlated with Notch signaling and mitotic activity, making it a helpful tool for selecting highly differentiated and functional hiPSC-ACMs and -VCMs (Fig. 6). Although this study did not elucidate the direct relationship between CD151 and the signaling

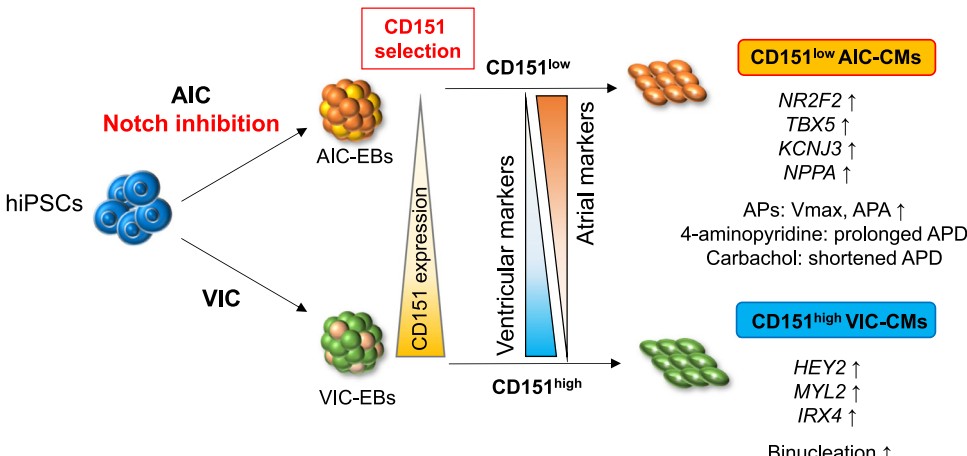

**Fig. 6 CD151 expression-based hiPSC-ACM and VCM purification.** A schematic diagram depicting our work to distinguish AIC-CMs and VIC-CMs derived from hiPSCs at days 20–30 based on CD151 expression. Under AIC, Notch inhibition is critical for generating ACMs with high efficiency.

pathways involved, we were able to obtain functional ACMs and VCMs from hiPSCs using a simple method, which represents a significant advance in the field. Furthermore, the mechanistic understanding of atrial differentiation from hiPSCs mediated by Notch signaling clarified here may help enhance the quality of ACMs during cardiac differentiation.

## Methods

**Human iPSC maintenance.** The hiPSC lines (TNNI1-EmGFP reporter, 1390C1, and 409B2) were established in our institute[23]. The TNNI1-EmGFP reporter and 1390C1 hiPSC lines were maintained on an iMatrix-511 (Nippi)-coated dish in AK02N medium (Ajinomoto) as previously described[23]. The 409B2 hiPSC line was maintained on SL10 feeder cells (REPROCELL) in Repro Stem medium (REPROCELL) supplemented with 5 ng/mL bFGF (REPROCELL) and penicillin/streptomycin (Sigma), as previously described[36]. Human iPSC studies were approved by the relevant ethical committee.

**Differentiation of hiPSCs into VCMs and ACMs.** To initiate cardiac differentiation of the TNNI1-EmGFP reporter and 1390C1 hiPSC lines, we used previously described protocols with some modifications[23,24]. Briefly, hiPSCs were dissociated into single cells with Accumax (STEMCELL TECHNOLOGIES), and EBs were generated using a 6-well ultra-low attachment plate at $2 \times 10^6$ cells/well in 1.5 mL/well StemPro-34 medium (Thermo Fisher Scientific) containing 2 mM L-glutamine (Sigma), 50 µg/mL ascorbic acid (AA, sigma), 0.4 mM monothioglycerol (MTG, Sigma), 150 µg/mL transferrin (Wako), 0.5% Matrigel (Corning), 10 µM ROCK inhibitor Y-27632 (Wako), and 2 ng/mL BMP4 (R&D). On day 1 of ACM differentiation, 1.5 mL StemPro-34 medium with the above supplements (without Y-27632 or Matrigel), 10 ng/mL bFGF (R&D, final 5 ng/mL), 4 ng/mL BMP4 (final 3 ng/mL), and 8 ng/mL Activin A (R&D, final 4 ng/mL) were added into the well for the AIC. For VCM differentiation, 1.5 mL StemPro-34 medium with the above supplements (without Y-27632 or Matrigel), 10 ng/mL bFGF (final 5 ng/mL), 18 ng/mL BMP4 (final 10 ng/mL), and 12 ng/mL Activin A (final 6 ng/mL) was added to the well to initiate VIC. On day 3, AIC-EBs were washed with IMDM (Thermo Fisher Scientific) once and then transferred to 3 mL StemPro-34 medium containing 2 mM L-glutamine, 50 µg/mL AA, 0.4 mM MTG, 150 µg/mL transferrin, 10 ng/mL VEGF (R&D Systems), 1 µM IWP-3 (Stemgent), 5.4 µM SB431542 (Wako), and 1 µM RA (Wako). VIC-EBs were washed once with IMDM and then transferred to 3 mL StemPro-

34 medium with the above supplements (without RA) and 0.6 µM Dorsomorphin (Sigma). On day 6, the medium was transferred to 2 mL StemPro-34 medium containing 2 mM L-glutamine, 50 µg/mL AA, 0.4 mM MTG, 150 µg/mL transferrin, and 5 ng/mL VEGF, and the EBs were maintained in this medium, with changes every 2–3 d until analysis. The plate was incubated at 37 °C in a hypoxia environment (5% $O_2$) for the first 10 d and then transferred to a normoxia environment.

For cardiac differentiation of the 409B2 hiPSC line, the hiPSCs were treated with dissociation solution for human ES/iPS cells (REPROCELL) for 5 min at 37 °C and then suspended at $5 \times 10^4$ cells/mL on the same day (0) with the above described medium. The single-cell suspension was seeded at 100 µL/well in a 96-well ultra-low attachment plate (Corning) for EB formation. On day 1, the same medium used above for AIC or VIC was added to each well at 100 µL/well. On day 3, EBs were dissociated using Accumax and then reaggregated with 3 mL StemPro-34 medium containing 2 mM L-glutamine, 50 µg/mL AA, 0.4 mM MTG, 150 µg/mL transferrin, 10 ng/mL VEGF, and 1 µM IWP-3 for VIC. AIC-EBs were washed once with IMDM and then transferred to 3 mL StemPro-34 medium containing 2 mM L-glutamine, 50 µg/mL AA, 0.4 mM MTG, 150 µg/mL transferrin, 10 ng/mL VEGF, 1 µM IWP-3, and 0.5 µM RA. On day 6, EBs were collected in 6-well ultra-low attachment plates and cultured for 4 weeks with the same medium and conditions described above.

**Monolayer differentiation into ACMs and VCMs.** Monolayer differentiation was performed according to a published protocol[37]. Briefly, hiPSCs were plated on Matrigel-coated 12-well multi-well plate. Two days after, 6 µM CHIR99021 was added to RPMI supplemented with B27 minus insulin (RPMI-ins, day 0). After 48 h (day 2), the medium was changed to RPMI-ins with 2 µM IWP-1. For AIC, on day3, 1 µM RA was added to the medium for 48 h. The medium was changed to RMPI-ins supplemented with 1 µM RA on day 5, RMPI-ins on day 6, and RPMI supplemented with B27 on day 8. For VIC, the medium was changed to RPMI-ins on day 5 and to RPMI + B27 on day 8. The medium was changed every 2 days during culture. Differentiated cells were used for analyses on days 20–23.

**Cell surface marker screening.** Day-20 AIC- and VIC-EBs were dissociated using Liberase and Accumax. Dissociated cells were assayed with the Lyoplate Human Cell Surface Marker Screening Panel (BD Biosciences) according to the manufacturer's

instructions. Flow cytometric analysis was performed using FACS Lyric (BD Biosciences).

**Flow cytometric analysis**. Differentiated EBs were dissociated in the same manner described in the cell surface marker screening section above. For TNNI1-EmGFP reporter hiPSC-CMs, purified mouse anti-human CD151 antibody (BD Biosciences, 1:200) was added as a primary antibody, and CD151 expression was detected using APC-anti-mouse IgG antibody (BD Biosciences, 1:200) or Alexa Fluor 647 goat anti-mouse IgG (Thermo Fisher Scientific, 1:200) as the secondary antibody. 1390C1 and 409B2 hiPSC-CMs were isolated using anti-SIRPa-PE/Cy7 (Biolegend, 1:500) as a cardiomyocyte marker and anti-CD90-APC (BD Biosciences, 1:2500), APC anti-human CD31 (Biolegend, 1:500), Alexa-Fluor647 anti-CD49a (Biolegend, 1:500), and APC anti-CD140b (Biolegend, 1:500) as non-cardiomyocyte lineage markers[24] and directly stained with PE anti-human CD151 (BD Biosciences, 1:200). For Ki-67 staining, fixed cells were stained with anti-human ACTN2 antibody (Creative Diagnostics, 1:200). The cells were then stained with Alexa Fluor 647 anti-mouse/human Ki-67 antibody (BioLegend, 1:400), PE anti-human CD151 (BD Biosciences, 1:200), and Alexa Fluor 488 donkey anti-rabbit IgG (Thermo Fisher Scientific, 1:200) as a secondary antibody against anti-human ACTN2 antibody. For binucleation analysis, dissociated cells were stained with anti-human ACTN2 antibody (Creative Diagnostics, 1:200), Alexa Fluor 488 donkey anti-rabbit IgG (Thermo Fisher Scientific, 1:200) as a secondary antibody, and Hoechst (DOJINDO, 1:1000). The cells were detected and sorted using FACS Aria Fusion (BD Biosciences) and analyzed using FlowJo software (v10.7.1). The gating strategies are shown in Fig. S8.

**Immunocytochemistry**. AIC- and VIC-CD151^high/low CMs were sorted and then seeded at $1 \times 10^5$ cells/well on a fibronectin-coated 96-well plate (Corning). The cells were fixed after 5–7 d of culture for 20 min with 4% paraformaldehyde, permeabilized for 15 min with 0.1% TritonX100-PBS, and then blocked for 1 h at room temperature with 2% goat serum/0.1% TritonX100-PBS. The fixed cells were stained using anti-MLC-2A (Synaptic systems, 1:100) and anti-MLC-2V (Proteintech, 1:200) overnight at 4 °C. The following day, the cells were washed twice with PBS and stained using Alexa Fluor 647 goat anti-mouse IgG (Thermo Fisher Scientific, 1:500), Alexa Fluor 647 goat anti-rabbit IgG (Thermo Fisher Scientific, 1:500), or Alexa Fluor 488 goat anti-rabbit IgG (Thermo Fisher Scientific, 1:500) as the secondary antibody for 1 h at 4 °C under dark conditions. The cells were washed twice with PBS and stained with Hoechst (DOJINDO) for 5 min at room temperature. Images were captured using a BZ-X710 (KEYENCE) with a 20× objective.

**Gene expression (qPCR)**. The total RNA content was prepared using a miRNeasy Micro Kit (QIAGEN), and purified RNA was reverse transcribed into cDNA using a SuperScript™ VILO™ cDNA Synthesis Kit (Thermo Fisher Scientific). Next, qPCR was performed on QuantStudio 7 Flex (Thermo Fisher Scientific) using Taqman probes (Applied Biosystems). The TaqMan probes used are listed (Supplementary Table 2).

**Patch-clamp electrophysiology**. AIC- and VIC-CD151^high/low CMs sorted by flow cytometry were plated on fibronectin-coated cover glasses (3 mm × 7 mm) in a single well of a 24-well plate ($2 \times 10^4$ cells). The CMs were cultured for 24 h in the same medium used on day 6 of cardiac differentiation described above with 10 μM Y-27632 and maintained in the same medium without Y-27632 with a medium change every 3 d until analysis.

The CMs were used for patch-clamp measurements 13–19 d after plating, using an Axopatch 200B amplifier (Molecular Devices) at 5 kHz in the current-clamp mode. APs of spontaneously beating single cells were recorded using the gap-free protocol, and signals were digitized at 10 kHz using Digidata 1322 A. The data were analyzed using pCLAMP 9.2 or 10.7 software (Molecular Devices) and an inverted microscope equipped with differential interface optics (Olympus). The CMs were perfused with Gey's buffer salt solution (Sigma) and kept at 35–37 °C in the chamber. Patch pipettes were made from glass capillaries using a micropipette puller (P-97/IVF, Sutter Instruments) with tip resistances of 3–5 MΩ when filled with the pipette solution. The pipette solution comprised 130 mM KOH, 20 mM KCl, 1 mM $MgCl_2$, 5 mM NaCl, 130 mM L-Aspartic acid, 10 mM HEPES, 10 mM EGTA, and 5 mM Mg-ATP and was adjusted to pH7.2 with KOH. APs were recorded and analyzed using Clampfit 10.7 software (Axon Instruments), and MDP, APA, APD, and Vmax were calculated based on the average AP of 10 consecutive and stable waves. APD was corrected using Bazett's correction[38].

**CD151 KO hiPSC line generation**. CD151 KO hiPSCs were generated from the TNNI1-EmGFP hiPSC line utilizing the CRISPR-Cas9 system. Gene editing was performed according to a published protocol[39], using a gRNA (target sequence: 5'-CAGGTTCCGACGCTCCTTGA-3'). Briefly, to form gRNA, an equimolar amount of crRNA and tracrRNA (IDT) was hybridized for 5 min at 95 °C and then cooled to 20 °C using a thermal cycler with a ramp rate setting of −0.1 °C/s. RNP complexes were formed with 61 pmol of gRNA and Cas9 nuclease each and transfected to TNNI1-EmGFP hiPSCs by electroporation. Transfected hiPSCs were cultured and analyzed by flow cytometry. CD151 KO hiPSCs were purified by sorting of CD151-negative populations. To validate them at the genome level, genomic DNA was isolated from TNNI1-EmGFP hiPSCs and CD151 KO hiPSCs. We then sequenced the regions around the target site of *CD151* and five off-target genes predicted by CRISPOR[40] using primers listed in Supplementary Table 3. The Sanger sequencing results were analyzed to identify the constitution of indels and the gene-editing efficiency using DECODR (https://decodr.org/).

**Western blots for CD151 expression**. Proteins were extracted from cells with RIPA buffer, with protein concentrations measured using the Pierce BCA Protein Assay Kit. Proteins were separated by SDS-PAGE using Criterion TGX Precast Gel (BioRad) and transferred to PVDF membrane. CD151 monoclonal antibody (2A8G8) (Thermo Fisher SCIENTIFIC, 1:2000) and anti-Actin monoclonal antibody (MAB1501) (Merck Millipore, 1:5000) were used as primary antibodies. ECL peroxidase-labeled anti-mouse antibody (NA931) (GE Healthcare, 1:5000) was used as the secondary antibody.

**Optical recording of APs with 4-AP and CCh**. AIC-CMs and LY411575/CD151^low ACMs were sorted on day 20 and plated as 5 μL cell suspensions ($5 \times 10^4$ cells) on a fibronectin-coated glass-bottom dish to form CM sheets, with the medium added 2 h later. The medium was changed every 2–3 d during CM culture. The CMs were treated with FluoVolt dye (Thermo Fisher Scientific) following the manufacturer's instructions. One hour after setting the cells in a stage top incubator (TOKAI HIT) at 37 °C (5% $CO_2$, 95% air), fluorescence measurements were acquired with an ECLIPSE Ti-E inverted fluorescence microscope (Nikon), objective Fluor 10×/0.45 (Nikon), X-Cite TURBO (Excelitas Technologies), EM-CCD camera ImagEM (Hamamatsu Photonics), and AQUACOSMOS 2.6 software (Hamamatsu Photonics) that

set the excitation wavelength at 490 nm and bandwidth at 10 nm. The X-Cite TURBO was set at 5% output. CMs cultured for 7 d were used to record the subarray images every 5 ms. Binning was set to 1 × 1. The regions of interest (ROIs) were defined as whole 512 × 64 pixels in the monolayers. The CMs were paced at 4 Hz with 2–5 ms depolarizing pulses at 10 V using a pulse stimulator (Master-9, A.M.P.I., Jerusalem, Israel) with an interelectrode distance of 12 mm (Intermedical, Osaka, Japan). The effects of drugs were measured 10 min after exposure to 50 μM 4-AP. Waveform traces were superimposed using OriginPro 2021 (OriginLab, Northampton, MA, USA). 4-AP (Sigma) was prepared as a 50 mM stock solution in Gey's buffer salt solution, with pH adjusted to 7.4, and stored at −20 °C. CCh (Sigma) was prepared as a 10 mM stock solution in Gey's buffer salt solution and stored at −20 °C.

**Library preparation and RNA-seq**. Total RNA was prepared with a miRNeasy Micro Kit. RNA-seq libraries were generated using TruSeq Stranded Total RNA (Illumina) according to the manufacturer's instructions. RNA-seq libraries were sequenced on a NextSeq 500 using High Output Kit v2.0 with single-end reads and 75 cycles (Illumina). RNA-seq reads were mapped to the human reference genome (hg38) using STAR[41] and normalized using DESeq2[42]. Genes with less than 10 normalized counts in an average of 12 samples were excluded, and 17142 genes were finally included in the data sets for the analysis. DEGs were identified using Wald's test in DESeq2. $P$ values of the genes were adjusted via the Benjamini-Hochberg procedure, and adjusted $p$ values < 0.05 were considered significant. PCA was performed using the prcomp package in R (v4.0.3) with the normalized gene expression data. GO and Reactome pathway enrichment analyses were performed using the geneXplain® platform (geneXplain) for DEGs. Hierarchical clustering was performed using Spearman's distance and Ward's algorithm, and heatmaps were generated using heatmap.2 in gplots (v3.1.1) R package.

**Single-cell RNA sequence analysis**. To assess the tissue-specific expression patterns of *CD151* in the developing human and mouse heart, we utilized publicly available single-cell RNA sequencing (scRNAseq) datasets[30,31] (Human: GSE106118, Mouse: GSE193346) across distinct developmental stages. For human samples, transcriptomic data from left atrial (LA) and ventricular (LV) tissues at 7, 13, and 20 gestational weeks were analyzed. Violin plots were generated at each stage to compare expression profiles, and statistical significance was evaluated using the Mann–Whitney $U$ test. A parallel methodology was applied to mouse samples, focusing on embryonic days 10.5 (E10.5) and 18 (E18), and postnatal day 5 (P5) with corresponding statistical analysis.

**Notch inhibitor treatment**. EBs were treated with 3 μM LY411575 (Selleck Chemicals), a gamma-secretase inhibitor, from days 8–20. On day 20, the EBs were dissociated and sorted in the same way as explained above in the flow cytometric analysis section.

**Statistics and Reproducibility**. GraphPad Prism (Graph Pad Software Inc.) was used to perform statistical analyses and create graphs and bar plots. Quantitative data are shown as the mean ± standard error of the mean (SEM), as indicated in the figure legends. Sample sizes (n) used in each experiment are indicated in the figure legends. Statistical significance was determined using Student's $t$ test (unpaired or paired, two-tailed) and one-way analysis of variance (ANOVA) followed by Tukey's honest significant difference (HSD) test or Dunnett's test, unless specified otherwise.

**Reporting summary**. Further information on research design is available in the Nature Portfolio Reporting Summary linked to this article.

## Data availability

The RNA-seq data reported in this paper have been deposited in NCBI's Gene Expression Omnibus and are accessible through GEO series accession number GSE179769. Source data are provided in Supplementary Data 3. The data that support the findings of this study are available from the corresponding author upon reasonable request.

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

## Acknowledgements
This work was supported by the Takeda-CiRA collaboration program and the following grants: JSPS KAKENHI Grants (JP18K15120, JP18KK0461, JP19K16041, JP17H04176, and JP 21H02912); the Leducq Foundation (18CVD05); the Research Center Network for Realization of Regenerative Medicine, AMED (JP20bm0104001, JP21bm0204003, JP21bm0804008, and JP21bm0804022); the Acceleration Program of R&D and implementation for Regenerative Medicine and Cell and Gene Therapy, AMED (JP23bm1423011 and JP23bm132300); the Research on Regulatory Science of Pharmaceuticals and Medical Devices, AMED (JP21mk0101189 and JP22mk0101241); the Translational Research grant, AMED (JP22ym0126091); the Research Project for Practical Applications of Regenerative Medicine, AMED (JP21bk0104095); and the iPS Cell Research Fund. We thank Shinya Yamanaka, Seigo Izumo, Yasushi Kajii, and Steve Okada for supporting this project; Takako Sono, Ayaka Sakoda, and Izumi Yamada (T-CiRA) for technical assistance; Kaoru Shimizu, Rumi Fujihara, and Mikako Marx-Mori (CiRA), Takanori Matsuo, Aya Higashide, and Yumi Monnai (T-CiRA) for their administrative support; and Peter Karagiannis and Kelvin Hui (CiRA) for critical reading of the manuscript. In preparing this work, ChatGPT was used to proofread the manuscript.

## Author contributions
M.N.K., K.M., A.H., K.I., T.N. and Y.Y. conceived and designed the study. M.N.K., Y.N., M.S., T.W., Misato.N., K.C. and S.C.N. performed the experimental work. M.N.K., M.S., C.O., Megumi. N. and R.H. performed and analyzed the RNA-seq experiments. M.N.K., K.M. and Y.Y. interpreted the data and wrote the manuscript. All authors discussed the results.

## Competing interests
The authors declare the following competing interests: Y.Y. is a scientific advisor of Orizuru Therapeutics, Inc. and received research funding from Takeda Pharmaceutical Company, Ltd. and Altos Labs, Inc. S.C.N. and K.I. are employees of Takeda Pharmaceutical Company, Ltd. T.N. is an employee of Orizuru Therapeutics, Inc. M.N.K, K.M., and Y.Y. are the inventors of the patent application (WO2021/033699 and 2022/125789). The other authors declare no competing interests.
