## [Peer Review File · Communications Biology]

Reviewers' comments:

Reviewer #1 (Remarks to the Author):

Despite considerable interest in generating cardiomyocytes from human pluripotent stem cells (hPSCs), it has remained challenging to generate pure populations of atrial-specific or ventricular-specific cardiomyocytes. Generation of specific cardiomyocyte subtypes is important for developmental biology, cardiac disease modeling, drug screening and other applications. While there exist protocols to bias hPSC differentiation towards atrial or ventricular cardiomyocytes, they typically fall short of generating homogeneous populations of either atrial or ventricular cardiomyocytes. In the present submission, Nakanishi-Koakutsu et al. introduce CD151 as a cell-surface marker that may help distinguish between these two cardiomyocyte subtypes: they claim that atrial and ventricular cardiomyocytes are CD151^{low} and CD151^{high}, respectively. They further show that NOTCH inhibition promotes certain aspects of atrial cardiomyocyte differentiation. Using electrophysiological criteria, they find that ventricular cardiomyocytes comprise ~93% of CD151^{high} populations, whereas atrial cardiomyocytes comprise ~80% of NOTCH inhibitor-treated CD151^{low} populations, within their differentiation system.

The introduction of CD151 as a cell-surface marker to distinguish hPSC-derived atrial vs. ventricular cardiomyocytes certainly represents an advance. However, several issues remain outstanding.

Major comments:

1. Population purity. If the authors claim that CD151 enables the isolation of pure atrial or ventricular cardiomyocytes ("These findings suggested that CD151^{low} ACMs and CD151^{high} VCMs were purified populations of ACMs and VCMs, respectively", pg. 7) they should isolate CD151^{low} vs. CD151^{high} populations by FACS and test whether atrial or ventricular markers are homogeneously expressed. What percentage of cells in CD151^{low} vs. CD151^{high} populations express atrial or ventricular markers using single-cell assays, such as flow cytometry, immunostaining, or single-cell RNA-seq? While the authors attempted to examine population homogeneity through electrophysiological measurements (Fig. 2c, Fig. 3h), analyses of marker expression will be critical. In particular, bulk-population qPCR assays in Fig. 1f,g show a roughly ~2-to-3-fold enrichment of atrial or ventricular markers after CD151 sorting, raising the question of how homogeneous the positively or negatively selected populations are. (For instance, NR2F1/2 do not show statistically significant expression differences between CD151^{high} and CD151^{low} atrial populations in Fig. 1f.)

2. Emphasizing the prior knowledge gap. We have a suggestion for the authors to better introduce the prior knowledge gap in the field, and how their work helps to fill this gap. As the authors point out, the published atrial and ventricular differentiation protocols generate heterogeneous cell populations containing some of the opposing cell-type (Fig. 1c). (For instance, "We confirmed that AIC- and VIC-EBs also contained MLC2v- and MLC2a-positive CMs, respectively, suggesting that AIC and VIC resulted in a mixture of VCM-like and ACM-like cells.") This is an extremely important point that should be emphasized in the Introduction, as the failure of extant differentiation protocols to generate pure atrial or ventricular cardiomyocytes provides an impetus for the present study. It will also help readers to understand why, in Fig. 1e, CD151-high and -low cells are both present in both atrial and ventricular differentiation protocols. To make it clearer, perhaps the authors should use the terminology "atrial- or ventricular-biased differentiation", as "atrial differentiation" and "ventricular differentiation" makes it sound like homogeneous cell populations are produced.

Minor comments:

1. Fig. 1b – for all FACS plots throughout this study, including Fig. 1b, population percentages should be shown
2. Fig. 1e – FACS gating controls (e.g., unstained cells, or marker-negative cells) should be shown so the reader knows how CD151 “high” vs. “low” were defined. At present, the cutoff between how CD151 “high” vs. “low” is not clearly defined.
3. Grammar: “Wnt signaling exerts a biphasic effect, early promotion, and late inhibition of cardiac development” and “In zebrafish, BMP signaling contributes to the cardiac mesoderm” (Introduction)
4. Missing citations: “These findings apply to hPSC-CMs, because manipulating BMP and Activin/Nodal signaling enhances the generation of hPSC-ACMs and -VCMs” (Introduction)
5. On pg. 5, TNNI1-EmGFP hPSCs are mentioned for the first time and should be briefly introduced. Do they carry a genomic knock-in or randomly integrated transgene? Also, if they have been published before, a reference should be cited.
6. In scRNAseq datasets of the developing human and mouse fetal heart across various embryonic stages, does CD151 show preferential expression in atrial vs. ventricular tissues?
7. Fig. 4a – Marker gene differences between CD151^{low} vs. CD151^{high} population are often quite modest.
8. Fig. 4b – In some circles, “2n” typically refers to 2n genome content, that is, a diploid cell. Would “4n” be appropriate?
9. “These data suggested that suppressing HEY2 by inhibiting Notch induced atrial-related genes to promote atrial specification.” – This claim should be toned down, if there is no direct data demonstrating that HEY2 loss alters differentiation. Claims that CD151 is a NOTCH target gene should also be toned down without direct evidence of NOTCH regulating the promoter and/or enhancer elements of CD151.
10. Fig. 3 – After NOTCH inhibition, does CD151 expression change in the bulk population? Put another way, do the overall percentages of atrial or ventricular cardiomyocytes change after NOTCH inhibition?
11. If NOTCH inhibition promotes atrial differentiation, does NOTCH activation enhance ventricular differentiation? This question might be out of the scope of the present study, but is worth mentioning.

Reviewer #2 (Remarks to the Author):

The manuscript by Nakanishi-Koakutsu and colleagues describes CD151 as a differentially expressed surface marker during cardiomyocyte differentiation applicable for enrichment of ventricular and atrial cardiomyocyte subtypes. Presumed identity of cardiomyocytes was investigated using gene expression analysis and electrophysiological measurements. Further, the authors identified Notch signaling being important during atrial differentiation. However, involvement of Notch signaling during cardiac differentiation is not novel. The close interaction between retinoic acid signaling and notch signaling during cardiomyocyte differentiation and atrial subtype specification had been reported previously (nicely reviewed here: <https://doi.org/10.1002/stem.3178>). Further, although the authors provide evidence for a correlation of CD151 expression and Notch signaling, the link between both does not become clear, particularly as Notch signaling inhibition does not alter CD151 expression. More critical, the very early time point of analysis around day 30 of differentiation (and probably before a robust functional/electrophysiological phenotype has developed) hampers my enthusiasm. Further, statistical analysis is a concern. Several concerns are particularly described below.

Major points:

1. The MLC2A and MLC2V analysis at day 27 appears rather early in CM differentiation, as ventricular CMs typically co-express both MLC2A and MLC2V until day 45-60. Did the authors check the heterogeneity of atrial/ventricular cultures at later time points of differentiation?
2. Does the surface marker expression change during differentiation and long-term culture or does the

differential marker expression remains constant? Is there a critical time-window to perform the CM selection? Analysis of CD151 expression at later time points of differentiation would significantly underpin the conclusions.

3. Could the authors explain why SSEA-4 (with its exclusive expression in VCMs?) was excluded from subsequent analyses? Why is high versus low (as for CD151) a superior selection strategy than high versus no (as for SSEA-4)?

4. Statistical analysis of gene expression data is a concern. Figure 1f/g, Figure 3b/d/g/i, Figure 4a/c (as well as various Suppl. Figures, e.g. S2, S3, S4, S6): Did the authors check their data for normality distribution before applying a parametric t-test? The low n-numbers suggest that a non-parametric test is more suitable. Further, as the authors are comparing four groups in multiple assays (e.g. Figure 1f/g), the parametric ANOVA or the non-parametric Kruskal-Wallis test should have been used. Overall, sample sizes appear rather low.

5. The n-numbers for patch clamp data (Figure 2) are rather low with 17 or less measured cells for each group from three differentiations. Please also show dot plots to better indicate variability between cells. More "mature" CMs around day 60 would be more appropriate for patch clamp experiments.

6. How do the authors explain the unexpectedly low percentage of atrial-like action potentials in the CD151^{low} ACM fraction? This is in huge contrast to previous reports of atrial diff protocols and also in contrast to the gene expression profiles and immunostainings in the previous figure (suggesting that the authors obtained almost homogenous cell populations). How do the ratios change in more mature CMs around day 60?

7. In line, the fraction of nodal cells is also rather high, as nodal cells are generally largely underrepresented (<1%) in ventricular and atrial-specific differentiations. How do the authors explain these findings? Did the authors analyze expression of nodal-specific genes?

8. As the authors claim that CD151^{high} AIC-CMs display a nodal-like phenotype (Figure 2c), would Notch activation during AIC differentiation result in predominantly nodal CMs? Additional experiments are needed to provide evidence for these observations.

9. In Figure 4i, the control group for comparison of optical action potential recordings between AIC-CMs (both CD151^{high} and CD151^{low}) versus LY-treated CD151^{low} does not seem to be accurately chosen. The AIC-CM CD151^{low} group would serve as a more suitable control here.

10. In line, action potential shape from LY-treated CMs appear critically different between electrophysiological and optical recordings (Figure 4f versus 4i). Please comment.

Minor points:

11. The labeling for the distinct differentiations throughout figures 2-5 is misleading. ACMs in the headings should be AIC-CMs and VCMs in the headings should be VIC-CMs.

12. Axis labeling needs to be corrected/harmonized, e.g. Suppl. Figure S2e/f CD151^{high} instead of VIC-high?

13. In the figure legends, the authors state the number of "independent experiments per group". Please define whether these are independent differentiations or different wells from the identical differentiation.

14. Suppl. Table 1 is only provided for VCMs, the ACM data is missing. Further, APD30/50/90 data for ACMs is interesting and should be included.

15. Abstract: "Furthermore, CD151^{low} ACMs differentiated via a Notch signaling inhibitor generated ACMs highly efficiently". 'ACMs generated ACMs efficiently' is misleading, please rephrase the sentence.

16. Line 122: "... using SIRPA and other lineage markers (Figs. S2a and S2d)". However, only SIRPA data is provided. Which other markers were utilized?

17. Line 156: "to elucidate the mechanism underlying the regulation of CD151 induced cardiac subtype differentiation...". This study did not examine whether the differentiation is actually induced by CD151, please rephrase.

Reviewer #3 (Remarks to the Author):

The manuscript entitled "CD151 expression marks atrial- and ventricular- differentiation from human induced pluripotent stem cells" submitted by Misato Nakanishi-Koakutsu et al. has identified a new cell surface marker CD151 for the identification of iPSC-derived cardiomyocytes that show atrial and ventricle-like gene expression and action potential. Their further study has found the expression of CD151 is correlated to Notch signaling in iPSC-CMs, and inhibition of Notch signaling promotes the differentiation of atrial-like iPSC-CMs. Although the finding is quite interesting, the lack of mechanistic understanding may limit its application in follow-up studies.

Overall, the manuscript presented a novel and potentially impactful work on the identification of a novel marker for atrial and ventricle-like iPSC-CM during the differentiation, which may attract many audiences in the field of stem cell research. However, since there are many concerns about the paper, I strongly recommend the authors to fix these issues through a major revision:

1: Figure 1e: the authors used the level of CD151 labeling in differentiating cells for the selection of atrial and ventricle-like iPSC-CM, yet it seems quite vague in the definition of the high/low expression of CD151 in this selection. As the signaling intensity of FACS may be affected by many factors, it will be important to clarify if the authors chose a certain CD151(APC or PE) value as the threshold, and what are the considerations for that specific cutoff.

2: the paper used EB-based iPSC-CM differentiation protocol. To validate if CD151 can act as a faithful marker, it will be helpful to test the CD151 another differentiation protocol, such as the 2d monolayer atrial and ventricle-CM differentiation protocols adapted from Xiaojun Lian et al. at 2013, which is known to be very efficient and were widely used in the field. Also, have the authors try to isolate the iCMs in a regulator differentiation (nor AIC or VIC) using CD151 as a marker?

3: it seems the differential expression of CD151 in AIC and VIC differentiation only last for a very short time window, have the authors followed the expression of the CD151 in the sorted ACM and VCM groups over time? Since the characterization of sorted ACM and VCM used only gene expression and patch clamp, would it be possible that the authors' classification of iCMs with CD151 only represent a same group of cells at different maturation stage? It would be great to see the isolated ACM and VCM cells maintain their atrial and ventricle-like CM phenotypes during prolonged culture: for example, if the authors compare the CD151-low and CD-151-high cells from both AIC and VIC groups at different time points after sorting, will they remain different and preserve the atrial and ventricle-like action potential and gene expression over time?

4: Figure 2 e, what is the criteria for ACM based on the AP recording?

5: CD151 was not regulated by Notch inhibition, yet the Notch4 and ligand expression is highlight related to CD151. Moreover, the ventricular marker genes are also highly correlated with CD151 expression. Would it be possible to testify if CD151 is an up-stream regulator of ACM and VCM related gene expression? Additional experimental evidence on CDC151-KO iPSC line will be necessary to find out more about the underlying mechanisms. Also, it will be really interesting to see what will be the composition of iPSC-CM types during differentiation without CDC151.

7: Notch inhibition lead to increased ACM gene expression, and AP change, increased MDP, why?

8: 4-ap are IKur blocker, have you also check the IK,ACh, another Atrial-specific current in CD151-low ACMs? What is the spontaneous beating rate of the ACM and VCM cells? When measuring the APD, have the authors used pacing (at different frequency, eg. 1~4 hz) to allow a fair comparison among different groups?

Reviewer #1 (Remarks to the Author):

Despite considerable interest in generating cardiomyocytes from human pluripotent stem cells (hPSCs), it has remained challenging to generate pure populations of atrial-specific or ventricular-specific cardiomyocytes. Generation of specific cardiomyocyte subtypes is important for developmental biology, cardiac disease modeling, drug screening and other applications. While there exist protocols to bias hPSC differentiation towards atrial or ventricular cardiomyocytes, they typically fall short of generating homogeneous populations of either atrial or ventricular cardiomyocytes. In the present submission, Nakanishi-Koakutsu et al. introduce CD151 as a cell-surface marker that may help distinguish between these two cardiomyocyte subtypes: they claim that atrial and ventricular cardiomyocytes are CD151low and CD151high, respectively. They further show that NOTCH inhibition promotes certain aspects of atrial cardiomyocyte differentiation. Using electrophysiological criteria, they find that ventricular cardiomyocytes comprise ~93% of CD151high populations, whereas atrial cardiomyocytes comprise ~80% of NOTCH inhibitor-treated CD151low populations, within their differentiation system.

The introduction of CD151 as a cell-surface marker to distinguish hPSC-derived atrial vs. ventricular cardiomyocytes certainly represents an advance. However, several issues remain outstanding.

Response:

We appreciate your helpful comments and suggestions. Our manuscript has been improved to make our claims clearer based on your advice. We have responded to your comments below and revised the manuscript.

Major comments:

1. Population purity. If the authors claim that CD151 enables the isolation of pure atrial or ventricular cardiomyocytes (“These findings suggested that CD151low ACMs and CD151high VCMs were purified populations of ACMs and VCMs, respectively”, pg. 7) they should isolate CD151low vs. CD151high populations by FACS and test whether atrial or ventricular markers are homogeneously expressed. What percentage of cells in CD151low vs. CD151high populations express atrial or ventricular markers using single-cell assays, such as flow cytometry, immunostaining, or single-cell RNA-seq? While the authors attempted to examine population homogeneity through electrophysiological measurements (Fig. 2c, Fig. 3h), analyses of marker expression will be critical. In particular, bulk-population qPCR assays in Fig. 1f,g show a roughly ~2-to-3-fold enrichment of atrial or ventricular markers after CD151 sorting, raising the question of how homogeneous the positively or negatively selected populations are. (For instance, NR2F1/2 do not show statistically significant expression differences between CD151high and CD151low atrial populations in Fig. 1f.)

Response:

Thank you for the critical suggestion to prove our claim. We analyzed the cell populations of MLC2a+ and MLC2v+ in CD151high and CD151low with AIC and VIC on day 20 by flow cytometer. As a result, in AIC, 75.8% and 15.7% of MLC2a+ cells in CD151low and CD151high population, respectively. In VIC, 44.6% and 17.2% of MLC2v+ cells in CD151high and

CD151^{low} populations, respectively. We further analyzed on day 60, 74.5% of MLC2^{v+} cells were detected in CD151^{high} population, and 58.6% of MLC2^v cells were detected in CD151^{low} population (Fig. 1g). Consistent with the electrophysiological data, these data indicated that atrial CMs with MLC2a expression were enriched in CD151^{low} population with AIC, and ventricular CMs with MLC2v expression were enriched in CD151^{high} with ViC. These results have been added to the manuscript in lines 149-151 (Figure 1g).

2. Emphasizing the prior knowledge gap. We have a suggestion for the authors to better introduce the prior knowledge gap in the field, and how their work helps to fill this gap. As the authors point out, the published atrial and ventricular differentiation protocols generate heterogeneous cell populations containing some of the opposing cell-type (Fig. 1c). (For instance, “We confirmed that AIC- and VIC-EBs also contained MLC2^{v-} and MLC2^{a+}-positive CMs, respectively, suggesting that AIC and VIC resulted in a mixture of VCM-like and ACM-like cells.”) This is an extremely important point that should be emphasized in the Introduction, as the failure of extant differentiation protocols to generate pure atrial or ventricular cardiomyocytes provides an impetus for the present study. It will also help readers to understand why, in Fig. 1e, CD151-high and -low cells are both present in both atrial and ventricular differentiation protocols. To make it clearer, perhaps the authors should use the terminology “atrial- or ventricular-biased differentiation”, as “atrial differentiation” and “ventricular differentiation” makes it sound like homogeneous cell populations are produced.

Response:

Thank you for the constructive suggestion. According to your comment, we employed the terminology involving atrial- or ventricular-biased differentiation. In addition, we have added sentences in lines 66-72 describing that ventricular CMs were still generated with atrial-biased differentiation and atrial-like CMs were generated with ventricular-biased differentiation.

Minor comments:

1. Fig. 1b – for all FACS plots throughout this study, including Fig. 1b, population percentages should be shown

Response:

Thank you for pointing this out. We added population percentages in all FACS plots.

2. Fig. 1e – FACS gating controls (e.g., unstained cells, or marker-negative cells) should be shown so the reader knows how CD151 “high” vs. “low” were defined. At present, the cutoff between how CD151 “high” vs. “low” is not clearly defined.

Response:

We apologize for the unclear explanation regarding the cutoff between CD151^{high} and low. We utilized unstained CMs with 2nd antibody alone as a negative control (NC), and the corresponding plots were incorporated into Fig. 1d. We defined the “CD151^{low}” population in a

manner such that >99% of this population falls within the gating of the NC sample, as detailed in the legend of Fig. 1d.

3. Grammar: *“Wnt signaling exerts a biphasic effect, early promotion, and late inhibition of cardiac development” and “In zebrafish, BMP signaling contributes to the cardiac mesoderm” (Introduction)*

Response:

We apologize for the grammatical mistakes. We have revised these sentences as follows:

“WNT signaling exerts a biphasic effect, promoting early cardiac development and inhibiting it later.”

“BMP signaling contributes to cardiac mesoderm differentiation in zebrafish, while Nodal signaling further promotes ventricular specification.”

4. Missing citations: *“These findings apply to hPSC-CMs, because manipulating BMP and Activin/Nodal signaling enhances the generation of hPSC-ACMs and -VCMs” (Introduction)*

Response:

Thank you for pointing this out. We have added the following citation to the manuscript in line 68.

S. J. Kattman, et al. *Cell Stem Cell* 2011 Vol. 8 Issue 2 Pages 228-40

5. On pg. 5, *TNNI1-EmGFP hPSCs* are mentioned for the first time and should be briefly introduced. Do they carry a genomic knock-in or randomly integrated transgene? Also, if they have been published before, a reference should be cited.

Response:

Thank you for the suggestion. We used the TNNI1-EmGFP/TNNI3-mCherry hiPSC line generated in our previous study (Nat Commun. 2021; 12: 3596.). This line was generated by placing EmGFP and mCherry elements downstream of TNNI1 and TNNI3, respectively, using CRISPR-Cas9. Since this study focuses on the EmGFP fluorescence to identify differentiated CMs, we labeled the cell line as “TNNI1-EmGFP hiPSCs”. We have added a brief explanation in the manuscript along with its citation (line 105-108).

6. In *scRNAseq* datasets of the developing human and mouse fetal heart across various embryonic stages, does *CD151* show preferential expression in atrial vs. ventricular tissues?

Response:

We analyzed CD151 expression of human and mouse fetal heart using publicly available scRNAseq data sets, as shown in Figure S7e. The analysis revealed that CD151 expression is significantly higher in left ventricular cells during early stages in both mice and humans.

However, the differential expression disappears in neonatal mouse hearts and late-stage human hearts.

Accordingly, we added the following sentences in lines 303-307.

“To investigate whether *in vivo* atrial and ventricular tissues show a similar difference in CD151 expression during development, we analyzed CD151 expression at the single-cell level in mice and human left atrium (LA) and ventricle (LV) using published data^{30, 31}. In line with our *in vitro* results, CD151 expressed higher in LV than in LA at mouse E10.5, human 7w, and 13w. However, such expression differences diminished during development (Fig.S7e).”

7. Fig. 4a – Marker gene differences between CD151^{low} vs. CD151^{high} population are often quite modest.

Response:

As you pointed out, there were modest differences in the expression levels of certain genes. Nonetheless, we observed significant differences in the expression levels of key maturation marker genes, such as MYH7 (1.7-folds) and MYL2 (2.4-folds), despite they were generated from the same differentiation culture batches. When considering the discrepancy in the percentage of binuclear cells, we concluded CD151^{high} VCMs are a population undergoing advanced VCM differentiation.

8. Fig. 4b – In some circles, “2n” typically refers to 2n genome content, that is, a diploid cell. Would “4n” be appropriate?

Response:

Thank you for the comment. We have revised “2x2n” instead of “2n” in Fig.4b to indicate binucleate cells

9. “These data suggested that suppressing HEY2 by inhibiting Notch induced atrial-related genes to promote atrial specification.” – This claim should be toned down, if there is no direct data demonstrating that HEY2 loss alters differentiation. Claims that CD151 is a NOTCH target gene should also be toned down without direct evidence of NOTCH regulating the promoter and/or enhancer elements of CD151.

Response:

Thank you for the suggestion. We have revised the manuscript to tone down the claim about the interaction between HEY2 expression and atrial specification in lines 222-224 as below.

“Although the causal role of HEY2 in atrial differentiation requires further investigation, our data highlight the potential involvement of HEY2 in mediating the gene expression changes downstream of Notch signaling, the inhibition of which promotes atrial specification.”

We also agree with your comment regarding the relationship between CD151 and Notch signaling. In this study, the ratio of CD151^{high} and low in LY411575 (Notch inhibitor) treated

AIC-CMs did not change compared to control AIC-CMs (Fig.S5f). This result suggested that CD151 was not regulated by Notch signaling. In lines 214-217, we added the sentence: “On day 20, the ratio of CD151^{high} ACMs to CD151^{low} ACMs remained unchanged with or without LY411575 treatment (Fig. S5f), suggesting that CD151 expression is not regulated by Notch signaling.”

10. Fig. 3 – After NOTCH inhibition, does CD151 expression change in the bulk population? Put another way, do the overall percentages of atrial or ventricular cardiomyocytes change after NOTCH inhibition?

Response:

As mentioned above, CD151 expression did not change after NOTCH inhibition (Fig. S5f). Nevertheless, we found that the percentage of atrial CMs was greater in NOTCH inhibition than in the control culture by patch-clamp experiment (e.g., 0% and 35% of atrial CMs in control AIC-CD151^{high} and CD151^{low}, respectively vs. 38% and 80% of atrial CMs in AIC-CD151^{high} and CD151^{low} with NOTCH inhibition, respectively). These results indicated that downregulated Notch signaling promoted atrial differentiation efficiency without altering the expression level of CD151. That also suggested that CD151 marks atrial differentiation independently of Notch signaling.

11. If NOTCH inhibition promotes atrial differentiation, does NOTCH activation enhance ventricular differentiation? This question might out of the scope of the present study, but is worth mentioning.

Response:

Thank you for the comment. We are also interested in investigating whether Notch activation promotes ventricular differentiation. However, it is worth noting that the contribution of Notch in ventricular differentiation may be limited, as NOTCH-related genes were expressed at low levels irrespective of CD151 expression level. Also, the previous report has shown that transient Notch activation in hPSC-derived cardiac mesoderm promotes immature cardiomyocytes with no change in their subtypes. While this experiment was not included at this time, we plan to explore it in future studies.

In lines 349-353, we added the sentence.

“A previous report showed that transient Notch activation in hPSC-derived cardiac mesoderm promotes immature cardiomyocytes with no change in their subtypes 34. Though the time points of Notch activation or inhibition are different between that report and our study, Notch signaling may be critical for regulating the maturation of both VCMs and ACMs rather than cell fate.”

In lines 360-362,

“In contrast, Notch activation after cell fate determination may not contribute to VCM differentiation efficiency since Notch signaling-related genes were expressed lowly in VIC-CMs (Figure S5d).”

Reviewer #2 (Remarks to the Author):

The manuscript by Nakanishi-Koakutsu and colleagues describes CD151 as a differentially expressed surface marker during cardiomyocyte differentiation applicable for enrichment of ventricular and atrial cardiomyocyte subtypes. Presumed identity of cardiomyocytes was investigated using gene expression analysis and electrophysiological measurements. Further, the authors identified Notch signaling being important during atrial differentiation. However, involvement of Notch signaling during cardiac differentiation is not novel. The close interaction between retinoic acid signaling and notch signaling during cardiomyocyte differentiation and atrial subtype specification had been reported previously (nicely reviewed here: <https://doi.org/10.1002/stem.3178>). Further, although the authors provide evidence for a correlation of CD151 expression and Notch signaling, the link between both does not become clear, particularly as Notch signaling inhibition does not alter CD151 expression. More critical, the very early time point of analysis around day 30 of differentiation (and probably before a robust functional/electrophysiological phenotype has developed) hampers my enthusiasm. Further, statistical analysis is a concern. Several concerns are particularly described below.

Response:

Thank you for your helpful comments and suggestions to improve our manuscript. We have added the data using day 60 of differentiation in addition to day 30 so far. Also, we have responded to your comments below and revised our manuscript as you suggested.

Major points:

1. The MLC2A and MLC2V analysis at day 27 appears rather early in CM differentiation, as ventricular CMs typically co-express both MLC2A and MLC2V until day 45-60. Did the authors checked the heterogeneity of atrial/ventricular cultures at later time points of differentiation?

Response:

Thank you for the comment. We checked MLC2a and MLC2v expression in atrial and ventricular cultures on day 60 by flow cytometric analysis. We found that day60 VCMs still expressed MLC2a in addition to MLC2v as you mentioned. On the other hand, MLC2v positive cells were a small population in atrial culture at the point of day20. These data are added in Supplementary Figure S1c to show the heterogeneity of conventional ventricular culture even at the later time of differentiation.

2. Does the surface marker expression change during differentiation and long-term culture or does the differential marker expression remains constant? Is there a critical time-window to perform the CM selection? Analysis of CD151 expression at later time points of differentiation would significantly underpin the conclusions.

Response:

Thank you for the valuable comment. The expression of CD151 gradually increases in cardiac differentiation (Fig. S7c). We have added the flow cytometry data showing the alteration in the

ratio between CD151^{high} and CD151^{low} TNNI+ cardiomyocytes from day 20 to day 60 in Fig. S7d. These results indicate that CD151 selection is effective between 20-30 days after differentiation but is not relevant for cardiomyocytes at later time points, such as day 60.

3. Could the authors explain why SSEA-4 (with its exclusive expression in VCMs?) was excluded from subsequent analyses? Why is high versus low (as for CD151) a superior selection strategy than high versus no (as for SSEA-4)?

Response:

This study aims to identify a marker that can selectively eliminate undesired subtypes in each subtype differentiation (e.g., ventricular- and nodal-like CMs in atrial differentiation and atrial- and nodal-like CMs in ventricular differentiation). We concluded that SSEA4 is not a suitable marker due to its absence in AIC-CMs, rendering it incapable of distinguishing any cells in AIC-CMs. Consequently, we have ruled out SSEA4 as a candidate marker.

4. Statistical analysis of gene expression data is a concern. Figure 1f/g, Figure 3b/d/g/i, Figure 4a/c (as well as various Suppl. Figures, e.g. S2, S3, S4, S6): Did the authors check their data for normality distribution before applying a parametric t-test? The low n-numbers suggest that a non-parametric test is more suitable. Further, as the authors are comparing four groups in multiple assays (e.g. Figure 1f/g), the parametric ANOVA or the non-parametric Kruskal-Wallis test should have been used. Overall, samples sizes appear rather low.

Response:

The normality of the data population was assessed using the Shapiro-Wilk test, and a parametric t-test was conducted subsequent to confirming a normally distributed dataset. To avoid the diminished power of the test, the parametric test was employed.

Regarding Fig. 1e, 1f, S2b, S2c, S2e, and S2f, our objective is not to conduct multiple testing with four samples, but to compare two groups: the CD151^{high} cells and their corresponding CD151^{low} cells. We, thus, did not employ multiple testing

5. The n-numbers for patch clamp data (Figure 2) are rather low with 17 or less measured cells for each group from three differentiations. Please also show dot plots to better indicate variability between cells. More “mature” CMs around day 60 would be more appropriate for patch clamp experiments.

Response:

Thank you for the suggestions. We have revised all the graphs displaying patch-clamp data into dot plots to show all data points.

We acknowledge your comment that using more mature CMs would be preferable for patch clamp experiments. Simultaneously, we observed the differences between CD151^{high} and CD151^{low} AIC-CMs (Figs. 2a and 2b) and between the AIC-CMs with a NOTCH inhibitor and those without one (Figs. 4f and 4g). Additionally, our AIC-CMs with Notch inhibition displayed

drug response to 4-AP and CCH (Figs. 4i and 4j). Based on these results, we are confident that our patch-clamp experiment was performed appropriately.

Regarding more mature CMs, we conducted gene expression analysis on day 60 cells. However, we did not observe any difference in the atrial marker genes between CD151^{high} and CD151^{low} AIC-CMs at this stage. Consequently, we decided to focus our patch-clamp experiments on CMs around day 30 rather than day 60 CMs.

As for the VIC-CMs, it is important to note that nearly all cells were in the CD151^{high} population on day 60 (Fig. S7d). Therefore, we believe that day 60 was not an appropriate time point to compare the electrophysiological phenotypes between CD151^{high} and CD151^{low}. Furthermore, we encountered experimental difficulties in isolating a sufficient quality of CD151^{low} cells for the assay.

6. How do the authors explain the unexpectedly low percentage of atrial-like action potentials in the CD151^{low} ACM fraction? This is in huge contrary to previous reports of atrial diff protocols and also in contrary to the gene expression profiles and immunostainings in the previous figure (suggesting that the authors obtained almost homogenous cell populations). How does the ratios change in more mature CMs around day 60?

Response:

Thank you for the comment. Although there was a low percentage with action potential of atrial CMs (APD_{30/90}<0.3 and V_{max}>10), we observed that CD151^{low} AIC-CMs had a significantly smaller APD_{30/90} ratio compared to CD151^{high} VIC-CMs (Fig.S3a). Additionally, the expression of atrial marker genes was higher in CD151^{low} AIC-CMs than in VIC-CMs, suggesting that CD151^{low} AIC-CMs were atrial-like CMs.

Due to the reasons mentioned in comment 5, we did not perform patch-clamp experiment on day 60. Since the expression of atrial markers did not change on day 20 and day 60, we anticipate the ratio of cells showing the atrial type action potential would not be increased on day 60.

7. In line, the fraction of nodal cells is also rather high, as nodal cells are generally largely underrepresented (<1%) in ventricular and atrial-specific differentiations. How do the authors explain these findings? Did the authors analyze expression of nodal-specific genes?

Response:

Thank you for the point out. Another study has reported showed similar percentages, around 5%, of nodal cells were detected in both ventricular and atrial differentiation (<https://www.cell.com/stem-cell-reports/pdfExtended/S2213-6711%2818%2930173-5>). The relatively high percentage of nodal-like cells can be attributed to the fact that immature cardiomyocytes often exhibit nodal-like phenotypes. We examined nodal-specific genes in AIC-CMs, as shown in Fig.S7h. While we observed 81% of nodal-like cells were present in CD151^{high} AIC-CMs (Fig. 2c), no significant differences in the expressions in the expression of nodal genes were detected between AIC-CD151^{high} and low CMs (Fig.S7i). Furthermore, we noticed that CD151^{high} AIC-CMs expressed atrial genes at lower levels compared to CD151^{low}-CMs. These findings suggest that nodal-like cells in our culture were not nodal cells but rather immature cells.

8. *As the authors claim that CD151^{high} AIC-CMs display a nodal-like phenotype (Figure 2c), would Notch activation during AIC differentiation result in predominantly nodal CMs? Additional experiments are needed to provide evidence for these observations.*

Response:

Thank you for the suggestion. Based on our findings, we believe that CD151^{high} AIC-CMs represent immature atrial-like cardiomyocytes rather than nodal cells, as discussed in our previous response.

We attempted Notch activation using a Notch activator compound, but it did not work in our system due to the technical difficulty. Furthermore, the previous report showed that transient Notch activation in hPSC-derived cardiac mesoderm promotes immature cardiomyocytes with no change in their subtypes. Although the time points of Notch activation is different from our inhibition time point, Notch signaling may be critical for regulating cell maturity rather than cell fate.

We added the sentences in lines 349-353,

“A previous report showed that transient Notch activation in hPSC-derived cardiac mesoderm promotes immature cardiomyocytes with no change in their subtypes¹. Though the time points of Notch activation or inhibition are different between that report and our study, Notch signaling may be critical for regulating the maturation of both VCMs and ACMs rather than cell fate.”

We are also interested in exploring the impact of Notch activation on ventricular and nodal differentiation as well as atrial differentiation and plan to establish a Notch activation system for further experiments.

9. *In Figure 4i, the control group for comparison of optical action potential recordings between AIC-CMs (both CD151^{high} and CD151^{low}) versus LY-treated CD151^{low} does not seem to be accurately chosen. The AIC-CM CD151^{low} group would serve as a more suitable control here.*

Response:

Thank you for the comment. In this context, our intention is to highlight the practical advantage of the combination method using CD151 selection and Notch inhibition in atrial differentiation, as opposed to primarily focusing on the effect on Notch inhibition. That is why we chose conventional AIC-CMs as a control. In lines 236-238, we have mentioned it in the manuscript as follows;

“To reveal the practical advantage of combining CD151 selection following Notch inhibition, we compared the responsiveness of LY411575/CD151^{low} ACMs and conventional AIC-CMs.”

10. *In line, action potential shape from LY-treated CMs appear critically different between electrophysiological and optical recordings (Figure 4f versus 4i). Please comment.*

Response:

Thank you for the point out. We think the differences were due to the utilization of data obtained from single-cell or monolayer CMs sheet. In a patch-clamp experiment, we employed single cells after cell sorting. On the other hand, monolayer CMs sheets were used for optical mapping assay to confirm the drug responses in cell sheets for future applications (e.g., disease models).

Minor points:

11. The labeling for the distinct differentiations throughout figures 2-5 is misleading. ACMs in the headings should be AIC-CMs and VCMs in the headings should be VIC-CMs.

Response:

Thank you for the suggestion. We revised the labeling as AIC-CMs and VIC-CMs in figures.

12. Axis labeling needs to be corrected/harmonized, e.g. Suppl. Figure S2e/f CD151^{high} instead of VIC-high?

Response:

Thank you for the point out and we apologize for making mistakes. We corrected appropriate labeling as same as other graphs.

13. In the figure legends, the authors state the number of “independent experiments per group”. Please define whether these are independent differentiations or different wells from the identical differentiation.

Response:

All experimental data were generated from independent differentiations. We revised figure legends as “n=x independent differentiation experiments per group”.

14. Suppl. Table 1 is only provided for VCMs, the ACM data is missing. Further, APD30/50/90 data for ACMs is interesting and should be included.

Response:

Thank you for the suggestion. We added the AIC-CMs data (APD30/90) in Supplementary Table 1 in addition to VIC-CMs data.

15. Abstract: “Furthermore, CD151^{low} ACMs differentiated via a Notch signaling inhibitor generated ACMs highly efficiently”. ‘ACMs generated ACMs efficiently’ is misleading, please rephrase the sentence.

Response:

Thank you for the point out. We revised the sentence in lines 40-42 as follows;

“Furthermore, Notch signaling inhibition followed by selecting the CD151^{low} population during atrial differentiation led to the highly efficient generation of ACMs as indicated by gene expression and electrophysiology.”

16. Line 122: “... using SIRPA and other lineage markers (Figs. S2a and S2d)”. However, only SIRPA data is provided. Which other markers were utilized?

Response:

We are sorry for the unclear explanation about the lineage markers. We used SIRPA and other lineage markers for gating CMs. The lineage markers were a mixture of four cell surface markers and shown as “lineage markers” in the Y-axis of the FACS plot (Fig. S2a and 2d). We added the word as “lineage markers: CD31, CD49a, CD90, and CD140b” in line 114.

17. Line 156: “to elucidate the mechanism underlying the regulation of CD151 induced cardiac subtype differentiation...”. This study did not examine whether the differentiation is actually induced by CD151, please rephrase.

Response:

Thank you for the suggestion. We revised the sentence in lines 192-193 as follows; “Next, we investigate whether CD151 expression is reflected by difference in signaling molecules and pathways regulating subtype.”

Reviewer #3 (Remarks to the Author):

The manuscript entitled “CD151 expression marks atrial- and ventricular- differentiation from human induced pluripotent stem cells” submitted by Misato Nakanishi-Koakutsu et al. has identified a new cell surface marker CD151 for the identification of iPSC-derived cardiomyocytes that show atrial and ventricle-like gene expression and action potential. Their further study has found the expression of CD151 is correlated to Notch signaling in iPSC-CMs, and inhibition of Notch signaling promotes the differentiation of atrial-like iPSC-CMs. Although the finding is quite interesting, the lack of mechanistic understanding my limited its application in follow-up studies.

Overall, the manuscript presented a novel and potentially impactful work on the identification of a novel marker for atrial and ventricle-like iPSC-CM during the differentiation, which may attract many audiences in the field of stem cell research. However, since there are many concerns about the paper, I strongly recommend the authors to fix these issues through a major revision:

Response:

We appreciate your comments and suggestions. We performed additional experiments to support our claims that CD151 works as a subtype selection marker. Also, we have responded to your comments below and revised the manuscript as you suggested.

1: Figure 1e: the authors used the level of CD151 labeling in differentiating cells for the selection of atrial and ventricle-like iPSC-CM, yet it seems quite vague in the definition of the high/low expression of CD151 in this selection. As the signaling intensity of FACS may be affected by many factors, it will be important to clarify if the authors chose a certain CD151(APC or PE) value as the threshold, and what are the considerations for that specific cutoff.

Response:

Thank you for your point out. In our analysis, CD151^{high/low} cutoff was defined comparing control (unstained sample or isotype control). We established the CD151^{high/low} gating in a way that nearly the entire CD151^{low} population (>99%) falls with the gating of negative control cells. Since this criterion was only explained in Fig.1d legend, we added the FACS plot of negative control to Fig.1d, S2a, and S2d.

2: the paper used EB-based iPSC-CM differentiation protocol. To validate if CD151 can act as a faithful marker, it will be helpful to test the CD151 another differentiation protocol, such as the 2d monolayer atrial and ventricle-CM differentiation protocols adapted from Xiaojun Lian et al. at 2013, which is known to be very efficient and were widely used in the field. Also, have the authors try to isolate the iCMs in a regulator differentiation (nor AIC or VIC) using CD151 as a marker?

Response:

Thank you for the suggestion. We incorporated CD151 selection into the monolayer differentiation protocol (STAR Protoc. 2020 Jun 3;1(1):100026.). As a result, similar to the EB method, the CD151^{low} population exhibited higher expressed atrial genes in atrial differentiation, and the CD151^{high} population demonstrated higher expressed ventricular genes in ventricular differentiation. These results highlight the efficiency of CD151 as a robust subtype selection marker. We added the data in the manuscript in the discussion (lines 290-293, Figs. S7a and S7b).

3: it seems the differential expression of CD151 in AIC and VIC differentiation only last for a very short time window, have the authors followed the expression of the CD151 in the sorted ACM and VCM groups over time? Since the characterization of sorted ACM and VCM used only gene expression and patch clamp, would it be possible that the authors' classification of iCMs with CD151 only represent a same group of cells at different maturation stage? It would be great to see the isolated ACM and VCM cells maintain their atrial and ventricle-like CM phenotypes during prolonged culture: for example, if the authors compare the CD151-low and CD-151-high cells from both AIC and VIC groups at different time points after sorting, will they remain different and preserve the atrial and ventricle-like action potential and gene expression over time?

Response:

In response to the reviewer's suggestion, we conducted culturing of CD151^{high} and low CMs after CD151 sorting on day 20 and evaluated subtype marker gene expression on day 60. The results demonstrated that the disparities vanished across all the genes in AIC (Fig S7f). This result potentially indicates that CD151 corresponds to atrial myocyte maturation in AIC, as pointed out by the reviewer. The capacity to select more matured cells at an early stage will hold significant promise for future applications in disease models. On the other hand, although the difference in MYL2 expression disappeared in VIC, differences in HEY2 and IRX4 persisted, suggesting that CD151 may not solely signify maturity in differentiating ventricular cells, but also the inherent nature of fully differentiated ventricular myocytes itself (high CD151 expression in VIC-CMs correlates with elevated expression of marker genes, signifying a greater resemblance to ventricular cells in the heart).

4: *Figure 2 e, what is the criteria for ACM based on the AP recording?*

Response:

We apologize for the unclear explanation about the criteria for the subtype-specific action potential. The criteria for ACM is $APD_{30/90} < 0.3$ and $V_{max} > 10$, as shown in Figure 2c. The description was added in the Figure 2e legend.

5: *CD151 was not regulated by Notch inhibition, yet the Notch4 and ligand expression is highlight related to CD151. Moreover, the ventricular marker genes are also highly correlated with CD151 expression. Would it be possible to testify if CD151 is an up-stream regulator of ACM and VCM related gene expression? Additional experimental evidence on CDC151-KO iPSC line will be necessary to find out more about the underlying mechanisms. Also, it will be really interesting to see what will be the composition of iPSC-CM types during differentiation without CDC151.*

Response:

Thank you for the suggestion. We established CD151 knockout iPSC line and investigated whether the expression of subtype markers would be affected or not. CD151KO hiPSC could differentiate into TNNI1⁺ CMs, and we observed no alteration in subtype marker genes expression between wild type and CD151 KO lines (Fig.3). These results indicated that CD151 did not govern the expression of atrial and ventricular genes. Additionally, we found that CD151 expression did not change by Notch inhibition that promoted atrial differentiation. Taken together, CD151 does not play a regulatory role in subtype differentiation but serves as a marker correlated with atrial/ventricular direction in cardiac differentiation. We have added this result in lines 179-189.

7: *Notch inhibition lead to increased ACM gene expression, and AP change, increased MDP, why?*

Response:

We found that Notch inhibition results in the downregulation of HEY2 expression (Fig. S5g). Given that HEY2 is recognized as a suppressor of atrial genes, we think that the repression of HEY2 due to the inhibition of the Notch pathway leads to the alleviation of atrial suppression. As a result, this inhibition contributes to the upregulation of atrial genes and a shift in action potentials toward an atrial profile. The increased MDP suggests the potential generation of more mature atrial cells, although the maturation status was not verified in this experiment. We have discussed the underlying mechanism in lines 357-360.

“These results indicate the inhibition of Notch signaling has the potential to promote atrial differentiation by reducing the expression of HEY2, a downstream target of Notch signaling and a repressor of atrial-related genes, such as TBX5, NPPA, and MYL7 25 (Fig. S5j).”

8: 4-ap are I_{Kur} blocker, have you also check the I_{K,ACh}, another Atrial-specific current in CD151-low ACMs? What is the spontaneous beating rate of the ACM and VCM cells? When measuring the APD, have the authors used pacing (at different frequency, eg. 1~4 hz) to allow a fair comparison among different groups?

Response:

Thank you for the suggestion. We performed optical imaging for CD151low ACMs using Carbachol (CCh), I_{K_{ACh}} agonist. We have incorporated these results in the manuscript in lines 232-242 and Fig.4j. As we expected, CD151low ACMs with Notch inhibition showed a response for CCh with shortening their action potential duration. However, conventional ACMs did not exhibit such a response (Fig.4j).

We apologize for omitting the information of the pacing rate in the initial submission. We performed the experiment under the pacing at 4Hz because the spontaneous beating rate was about 3Hz for control ACMs and CD151low ACMs with Notch inhibition. We added this pacing rate detail in the legend for Fig. 4j.

REVIEWERS' COMMENTS:

Reviewer #2 (Remarks to the Author):

The authors have adequately addressed many of the concerns raised by the reviewers.

Reviewer #3 (Remarks to the Author):

The authors have tried to address the comments with additional experiments during revision, which has improved the quality and credibility of the current work.

I have some additional minor concerns regarding the point-to-point response to my comments:

1: Regarding the gating of CD151 high/low populations in FACS, the authors used a cutoff at 99% of unstained CMs. However, it seems the unstained CMs also showed 2 populations in the CD151 channel, similar to the stained population. Please comment.

2: In the CD151 KO experiment, the authors applied the CRISPR-Cas9 tool to induce the knockout of CD151. However, the current work used FACS to sort out the iPSC population with low CD151 expression, which is quite tricky as A: about 71% of transfected cells were overlapped with isotype control, while the KO efficiency by RNP method could be much lower than that, so the FACS may only sort out a subpopulation of cells with an intrinsic low CD151 expression level instead of true KO. B: the cells in the sorted population are likely to be highly heterogeneous, which may lead to the authors' observation that CD151 has no effect on iPSC proliferation and differentiation. To make the conclusion more solid, the authors need to provide a genotyping sequence result on the target region of the sgRNA and confirm the indels have led to truncated CD151 on both genomic alleles. Moreover, genotyping results should also be provided for the potential off-target genomic loci of the sgRNA they used.

Reviewer #4 (Remarks to the Author):

The authors replied satisfactorily to all reviewer's comments.

Reviewer #2 (Remarks to the Author):

The authors have adequately addressed many of the concerns raised by the reviewers.

Response:

We greatly appreciate your positive comments.

Reviewer #3 (Remarks to the Author):

The authors have tried to address the comments with additional experiments during revision, which has improved the quality and credibility of the current work. I have some additional minor concerns regarding the point-to-point response to my comments:

Response:

Thank you for reviewing our manuscript again and providing your comments. We have addressed each of your comments below.

1: Regarding the gating of CD151 high/low populations in FACS, the authors used a cutoff at 99% of unstained CMs. However, it seems the unstained CMs also showed 2 populations in the CD151 channel, similar to the stained population. Please comment.

We apologize for presenting the data that raised concerns. Initially, unstained CMs appeared to consist of two populations due to the presence of cells exhibiting negative fluorescence in the CD151-APC channel when using a logarithmic scale for the CD151 axis. This issue was caused by fluorescent baseline subtraction error. By applying a biexponential transformation, we resolved this visual discrepancy. Consequently, we have revised the plots in Fig1d to reflect the biexponential scale. As a result, unstained CMs were now accurately represented as a single CD151-negative population on the plot.

2: In the CD151 KO experiment, the authors applied the CRISPR-Cas9 tool to induce the knockout of CD151. However, the current work used FACS to sort out the iPSC population with low CD151 expression, which is quite tricky as A: about 71% of transfected cells were overlapped with isotype control, while the KO efficiency by RNP method could be much lower than that, so the FACS may only sort out a subpopulation of cells with an intrinsic low CD151 expression level instead of true KO. B: the cells in the sorted population are likely to be highly heterogeneous, which may lead to the authors' observation that CD151 has no effect on iPSC proliferation and differentiation. To make the conclusion more solid, the authors need to provide a genotyping sequence result on the target region of the sgRNA and confirm the indels have led to truncated CD151 on both genomic alleles. Moreover, genotyping results should also be provided for the potential off-target genomic loci of the sgRNA they used.

A: The plot data plotted in the left panel of Fig. 3a show only 7% of WT (TNNI1-EmGFP) hiPSCs overlapped with isotype control. This suggests that 7% of WT hiPSCs exhibited intrinsically low CD151 expression, while the majority (approximately 93%) expressed CD151

at levels sufficient for separation. These results suggest that only a small population within our sorted cells had intrinsically low CD151 expression, indicating that our sorting method successfully isolated true KO cells.

B: We performed Sanger sequencing on the sgRNA target region of CD151 and on five predicted off-target genes. The sequencing data were analyzed to identify the constitution of indels and to determine the gene-editing efficiency using DECODR (<https://decodr.org/>). The results demonstrated that a frameshift was induced with 100% efficiency in CD151 KO hiPSCs and that no mutations were found at the predicted off-target sites in exons of each of the five genes. We have added those results in the manuscript in lines 181-183 and Figs. S4b and S4c.

Reviewer #4 (Remarks to the Author):

The authors replied satisfactorily to all reviewer's comments.

Response:

We greatly appreciate your positive comments.